# CTRL: Graph Condensation via Crafting Rational Trajectory Matching

## Abstract

Training on large-scale graphs has achieved remarkable results in graph representation learning, but its cost and storage have raised growing concerns. Generally, existing graph distillation methods address these issues by employing gradient matching, but these strategies primarily emphasize matching directions of the gradients. We empirically demonstrate this can result in deviations in the matching trajectories and disparities in the frequency distribution. Accordingly, we propose **CrafTing RationaL** trajectory (**CTRL**), a novel graph dataset distillation method. CTRL introduces gradient magnitude matching during the gradient matching process by incorporating the Euclidean distance into the criterion. Additionally, to avoid the disregard for the evenness of initial feature distribution that the naive random sampling initialization may introduce, we adopt a simple initialization strategy that ensures evenly distributed features. CTRL not only achieves state-of-the-art performances in 34 cases of experiments on 12 datasets with lossless performances on 5 datasets but can also be easily integrated into other graph distillation methods based on gradient matching. The code will be made public.

## 1 Introduction

Graph neural networks (GNNs) have demonstrated superior performances in various graph analysis tasks, including node classification (Li & Pi, 2019; Xiao et al., 2022), link prediction (Chen et al., 2022b; Rossi et al., 2022), and graph generation (Van Assche et al., 2022; Vignac et al., 2022). However, as deep neural networks, training GNNs on large-scale graph datasets comes at a high cost (Wang et al., 2022; Polisetty et al., 2023). One of the most straightforward ideas is to reduce the redundancy of the large graph. For example, graph sparsification (Li et al., 2022b; Yu et al., 2022b) directly removes insignificant edges, while graph coarsening (Chen et al., 2022a; Kammer & Meintrup, 2022) reduces the graph size by merging similar nodes. Yet the rigid operation graph sparsification and coarsening conduct heavily rely on heuristic techniques (Cai et al., 2021), which leads to the disregard for local details and limitation in condensation ratio (Jin et al., 2022b).

The recently proposed methods of graph dataset distillation generate a synthetic graph dataset to approximate the original graph dataset (Jin et al., 2022b;a; Zheng et al., 2023), which demonstrate advantages over traditional approaches due to their consideration of global information in graph data. As shown in Figure 1(a), GCond follows the vision dataset condensation (Zhao et al., 2020) and proposes a graph condensation method based on gradient-matching. It optimizes synthetic data by minimizing the gradient matching loss between the gradients of training losses w.r.t the GNN trained on the original graph and the synthetic graph. GCond is capable of condensing the graph with an extremely low condensation ratio, while the synthetic graph dataset is expected to achieve comparable results as training on the original graph dataset.

Although graph condensation methods based on gradient matching have achieved remarkable performance (Jin et al., 2022b;a), the criterion in gradient matching and the initialization they adopt are still naive. To elaborate, firstly, relying solely on cosine distance as a matching criterion (Jin et al., 2022b) disregards gradient magnitude, which will cause cumulative errors before achieving alignment between gradients of real and synthetic data. As shown in Figure 3(a) and 3(b), even if the cosine distance converges in later stages of matching, it fails to eliminate the accumulated errors caused by the lack of alignment in gradient magnitude. Secondly, random sampling for initialization

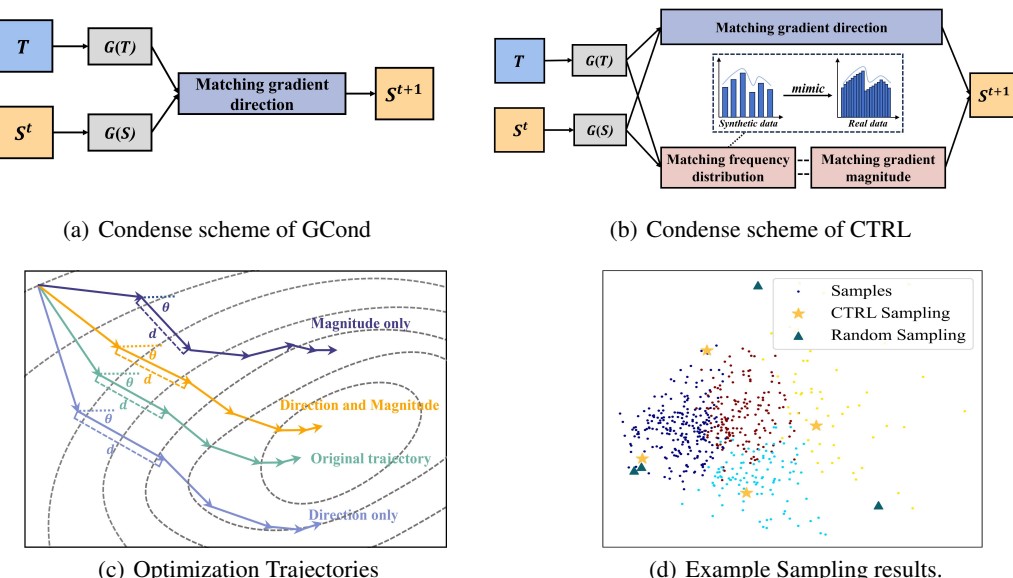

(a) Condense scheme of GCond        (b) Condense scheme of CTRL

(c) Optimization Trajectories        (d) Example Sampling results.

Figure 1: (a) and (b) illustrate the pipelines of GCond and CTRL. (c) presents a comparison of optimization trajectories when training from scratch using real data versus synthetic data generated under three distinct gradient matching strategies. (d) illustrates the disparity between CTRL sampling and random sampling in terms of selecting initialization values for synthetic data.

may result in synthetic data that initially exhibit similar features, as shown in Figure 1(d), thereby causing additional challenges in the optimization process (Liu et al., 2023).

In this paper, we introduce a novel graph condensation method called **C**raf**T**ing **R**ationa**L** trajectory matching (CTRL). CTRL considers both direction and magnitude of gradient (Cantrijn et al., 1992; Fukuda & Drummond, 2011; Ruder, 2016) during gradient matching. Aiming to avoid singular features, we cluster each class of the original data into sub-clusters and then samples from each sub-cluster as initialization values to cover the sample space of the whole class (Liu et al., 2023). Compared to previous methods, as shown in Figure 1(c) and Figure 1(d), CTRL allows a finer matching of gradients and obtains initial synthetic data with even feature distribution, effectively reducing the cumulative errors caused by misalignment in gradient magnitude and gradient direction (Figure 3(c) and 3(d)). Moreover, by matching the gradient magnitude, CTRL can effectively capture the frequency distribution of signals in the original graph, empirical studies (Table 6) reveal that the graph condensed by CTRL accurately mimics the frequency distribution of the original graph.

We further assess the performance of CTRL through comprehensive experiments, to evaluate its generalization capabilities, we extend its application to node classification and graph classification tasks. With greater specificity, CTRL effectively reduces the Euclidean distance between gradients to just 3.7% of their original values while maintaining the cosine distance. Remarkably, we achieve lossless results on prominent datasets, including Citeseer, Cora, and Flickr for node classification, as well as the Ogbg-molbace and Ogbg-molbbbp datasets for graph classification. Notably, our approach yields new state-of-the-art results on the Ogbg-molbace and Ogbn-arxiv datasets, demonstrating improvements of 6.2% and 6.3%, respectively. Our main contributions are summarized as:

- Based on the analysis of gradient matching, we introduce CTRL, a simple and highly generalizable method for graph dataset distillation through finer-grained gradient matching.

- In CTRL, the optimization trajectory of our synthetic data closely approximates that of real data through a weighted combination of cosine distance and Euclidean distance and effectively captures the feature distribution of real data.

- We conduct experimental evaluations across 18 node classification tasks and 18 graph classification tasks. The results highlight that our method achieves state-of-the-art performances in 34 cases of experiments on 12 datasets with lossless performances on 5 datasets.

## 2 How CTRL Achieves Effective Gradient Matching

### 2.1 Graph condensation via gradient matching

The objective of the graph condensation via gradient matching framework is to extract latent information from a large graph dataset $\mathcal{T} = \{\mathbf{A}, \mathbf{X}, \mathbf{Y}\}$ to synthesize a smaller dataset $\mathcal{S} = \{\mathbf{A}', \mathbf{X}', \mathbf{Y}'\}$, such that a model trained on $\mathcal{S}$ can achieve comparable results to one trained on $\mathcal{T}$, where $\mathbf{A} \in \mathbb{R}^{N \times N}$ denotes the adjacency matrix, $\mathbf{X} \in \mathbb{R}^{N \times d}$ is the feature, and $\mathbf{Y} \in \mathbb{R}^{N \times 1}$ represents the labels, the first dimension of $\mathbf{A}', \mathbf{X}'$, and $\mathbf{Y}'$ are $\mathbf{N}'$. To summarize, the process essentially revolves around matching the gradients generated during the training of Graph Neural Networks on both datasets.

To achieve this alignment, the following steps are undertaken: Fitstly, both on the large graph dataset $\mathcal{T}$ and the small synthetic dataset $\mathcal{S}$, we train a GNN model parameterized with $\boldsymbol{\theta}$, denotes as $\text{GNN}_{\boldsymbol{\theta}}$ and compute the parameter gradients at each layer for this model. We then use the gradients from the synthetic dataset to update the GNN model. The optimization goal is to minimize the distance $\mathcal{D}$ between the gradients at each layer, essentially, this aligns the training trajectories of the models over time, making them converge.

The above process can be described by the following formula:

$$\min_{\mathcal{S}} \mathcal{L}\left(\text{GNN}_{\boldsymbol{\theta}_{\mathcal{S}}}(\mathbf{A}, \mathbf{X}), \mathbf{Y}\right) \quad \text{s.t} \quad \theta_{\mathcal{S}} = \arg\min_{\boldsymbol{\theta}} \mathcal{L}(\text{GNN}_{\boldsymbol{\theta}}(\mathbf{A}', \mathbf{X}'), \mathbf{Y}'), \tag{1}$$

$$\min_{\mathcal{S}} \mathbb{E}_{\boldsymbol{\theta}_0 \sim P_{\theta_0}}\left[\sum_{t=0}^{T-1} \mathbf{D}\left(\nabla_{\boldsymbol{\theta}} \mathcal{L}\left(\text{GNN}_{\boldsymbol{\theta}_t^{\mathcal{S}}}(\mathbf{A}', \mathbf{X}'), \mathbf{Y}'\right), \nabla_{\boldsymbol{\theta}} \mathcal{L}\left(\text{GNN}_{\boldsymbol{\theta}_t^{\mathcal{T}}}(\mathbf{A}, \mathbf{X}), \mathbf{Y}\right)\right)\right], \tag{2}$$

where $T$ is the number of steps of the whole training trajectory, and $\boldsymbol{\theta}_t^{\mathcal{S}}, \boldsymbol{\theta}_t^{\mathcal{T}}$ denote the model parameters trained on $\mathcal{S}$ and $\mathcal{T}$ at time step $t$. The distance $\mathbf{D}$ is further defined as the sum of the distance between two gradients at each layer.

### 2.2 Graph condensation via crafting rational trajectory

**Weakness of cosine distance only.** In previous gradient matching-based graph dataset condensation methods (Jin et al., 2022b;a), the cosine distance was primarily used as a criterion to measure the distance between gradients. However, cosine distance mainly reflects the similarity in direction between gradients but can not effectively address the vector nature of gradients, *i.e.*, does not capture the differences in magnitude between gradients (Fukuda & Drummond, 2011; Ruder, 2016), which leads to bias in gradient matching. In the initial stages, due to the lack of convergence in matching gradients' magnitude and direction, the errors accumulate, resulting in significant trajectory deviations and ultimately a misalignment between the synthesized data and real data.

**Refined matching criterion.** To provide better guidance for gradient matching, inspired by Jiang et al. (2022), we introduce the Euclidean distance as a supplement to matching loss. Specifically, we utilize a linear combination of cosine similarity complement and Euclidean distance as the metric for measuring gradient directions and magnitudes. We assign different weights to balance the significance of these two metrics through a hyperparameter $\beta$. Given two gradient $\mathbf{G}^{\mathcal{S}} \in \mathbb{R}^{d_1 \times d_2}$ and $\mathbf{G}^{\mathcal{T}} \in \mathbb{R}^{d_1 \times d_2}$ at a specific layer, we defined the distance $\text{dis}(\cdot, \cdot)$ as follows:

$$\text{dis}(\mathbf{G}^{\mathcal{S}}, \mathbf{G}^{\mathcal{T}}) = \sum_{i=1}^{d_2} \left[(1-\beta)^* \left(1 - \frac{\mathbf{G_i}^{\mathcal{S}} \cdot \mathbf{G_i}^{\mathcal{T}}}{\left\|\mathbf{G_i}^{\mathcal{S}}\right\| \left\|\mathbf{G_i}^{\mathcal{T}}\right\|}\right) + \beta^*(\| \mathbf{G_i}^{\mathcal{S}} - \mathbf{G_i}^{\mathcal{T}} \|)\right], \tag{3}$$

where $\mathbf{G_i}^{\mathcal{S}}, \mathbf{G_i}^{\mathcal{T}}$ are the i-th column vectors of the gradient matrices. With the above formulation, we achieve vector gradient matching.

**Initialization of synthetic data.** Traditional graph condensation methods suffer from limitations in effectively capturing the feature distribution of real data. As shown in Figure 1(d), random sampling initialization brings uneven feature distribution. This leads to suboptimal initial optimization points in the distillation process, thereby causing a degradation of training efficiency. To tackle this problem, CTRL adopts K-Means (Hartigan & Wong, 1979; Arthur & Vassilvitskii, 2007) on the node features in each class of real data, dividing each class of real data into $M$ sub-clusters, where $M$ is the number

of synthetic data instances for that class. For each sub-cluster, we randomly sample a node feature from it to serve as the initial value for synthetic data. Such that the feature distribution of our synthetic data initialization becomes even closer to the feature distribution of real data (Liu et al., 2023), while it only requires clustering data from a specific class at a time, which is exceptionally cost-effective in terms of computation.

## 2.3 BRIDGING GRAPH FREQUENCY DISTRIBUTION TO GRADIENT MAGNITUDES

In this section, we initially introduce the definition and measurement methods of graph signal distribution as explored in prior research. Subsequently, we empirically demonstrate a strong correlation between graph frequency distribution and the gradient magnitudes generated during the training of GNNs. This observation further elucidates why incorporating gradient magnitude matching can yield improved results in the domain of graph dataset distillation.

**Measurement of graphs signals.** Graph spectral analysis provides insights into the frequency content of signals on graphs. This allows the graph signal to be decomposed into components of different frequencies based on the graph discrete Fourier transform (GDFT) (Stankovic et al., 2019; Cheng et al., 2022), as shown in Eq. (4) and Eq. (5).

$$L = I - D^{-\frac{1}{2}} A D^{-\frac{1}{2}} \quad = U \Lambda U^T, \tag{4}$$

$$\hat{x} = U^T x, \quad \hat{x}_H = U_H^T x, \quad \hat{x}_L = U_L^T x, \tag{5}$$

where $L$ denotes a regularized graph Laplacian matrix, $U$ is a matrix composed of the eigenvectors of $L$. $\Lambda$ is a diagonal matrix with the eigenvalues of $L$ on its diagonal, and $x = (x_1, x_2, \cdots, x_N)^T \in \mathbb{R}^N$ and $\hat{x} = (\hat{x}_1, \hat{x}_2, \cdots, \hat{x}_N)^T = U^T x \in \mathbb{R}^N$ represent a signal on the graph and the GDFT of $x$, respectively, while $U_H$ and $U_L$ correspond to matrices containing eigenvectors associated with eigenvalues that are above and below the specified cutoff frequency $\tau$. Furthermore, $\hat{x}_H$ and $\hat{x}_L$ represent signals attributed to high-frequency and low-frequency components respectively.

However, this process is often time-consuming due to the Laplacian matrix decomposition required. Following the previous work (Tang et al., 2022), we introduce the definition of the high-frequency area, represented by Eq. (6).

$$S_{\text{high}} = \frac{\sum_{k=1}^{N} \lambda_k \hat{x}_k^2}{\sum_{k=1}^{N} \hat{x}_k^2} = \frac{x^T L x}{x^T x}, \tag{6}$$

where $\hat{x}_k^2 / \Sigma_{i=1}^N \hat{x}_i^2$ denotes the spectral energy distribution at $\lambda_k (1 \le k \le N)$, as the spectral energy on low frequencies contributes less on $S_{\text{high}}$, this indicator increases when spectral energy shifts to larger eigenvalues. Note that $S_{\text{high}}$ is defined under the premise of $x$ being a one-dimensional vector. In subsequent computations, $S_{\text{high}}$ is computed for each feature dimension and averaged.

**Matching frequency distribution.** Previous studies have shown that aligning the frequency distribution of synthetic data with real data can enhance the quality of synthetic data (Martinkus et al., 2022; Luo et al., 2022). However, matching the frequency domain distribution during gradient matching is challenging due to time-consuming matrix decomposition and significant differences in the number of nodes between original and synthetic graphs. Surprisingly, through experiments with three graph models and four commonly used GNN models, we discovered a strong correlation between the frequency distribution and the gradient magnitude during training.

As shown in Figure 2, with the high-frequency area increasing gradually, the gradient size generated has an obvious upward trend. Furthermore, 8 of the 12 experiments showed a correlation greater than 0.95, more details of the experiment are in Appendix B.4. Based on these experimental results, we can further validate the effectiveness of our approach. Considering gradient magnitude alignment during the gradient matching process not only explicitly accounts for the vector properties of gradients, enhancing the precision of gradient alignment, but also indirectly assists in emulating frequency distribution similar to real data within synthetic data. We provide further evidence in Sec. 3.3.

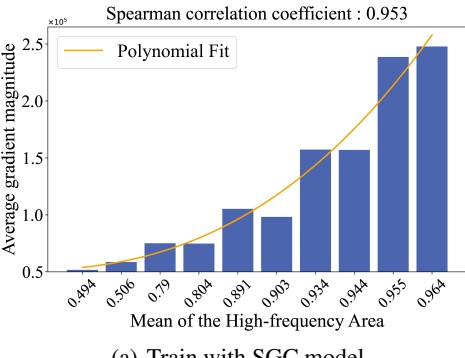 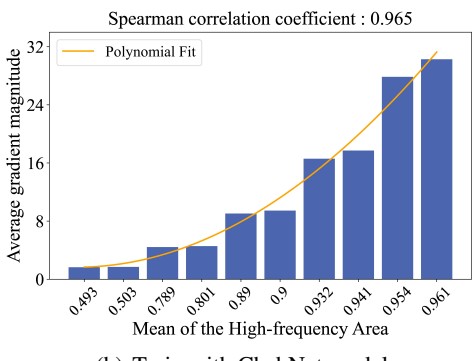

(a) Train with SGC model                (b) Train with ChebNet model

Figure 2: (a) and (b) demonstrate the relationship between average gradient magnitude and graph spectral distribution for synthetic graphs conforming to the Erdos-Renyi model, training with SGC and ChebNet respectively. The curves shown are the result of fitting a fourth-order polynomial. Altering the frequency distribution causes a significant impact on the model gradient during training.

## 3 EXPERIMENTS

### 3.1 DATASETS AND IMPLEMENTATION DETAILS

**Datasets.** To better evaluate the performance of CTRL, we conducted experiments on six node classification datasets: Cora, Citeseer (Kipf & Welling, 2016), Ogbn-arxiv (Hu et al., 2020), Ogbn-XRT (Chien et al., 2021), Flickr (Zeng et al., 2020), and Reddit (Hamilton et al., 2017), as well as six graph classification datasets, multiple molecular datasets from Open Graph Benchmark (OGB) (Hu et al., 2020), TU Datasets (MUTAG and NCI1) (Morris et al., 2020), and one superpixel dataset CIFAR10 (Dwivedi et al., 2020). For specific settings, we follow the settings of Jin et al. (2022b) and Jin et al. (2022a). More statistics on datasets are provided in Appendix A.1.

**Implementation details.** For node classification, without specific designation, in the condense stage, we adopt the 2-layer GCN with 128 hidden units as the backbone, and we adopt the settings on Jin et al. (2022b) and Jin et al. (2022a). For graph classification, we adopt a 3-layer GCN with 128 hidden units as the model for one-step gradient matching. Additionally, we finetuned our hyperparameter $\beta$ for different datasets. More details can be found in Appendix A.2.

**Evaluation.** We first learn synthetic graphs according to each type of baseline algorithm, then train a GCN classifier on which. Subsequently, we evaluate its classification performance on the real graphs' test sets. In the context of node classification tasks, we condense the entire graph and the training set graphs for transductive datasets and inductive datasets, respectively. For graph condensation, we repeat the generation process of condensed graphs 5 times with different random seeds and train GCN on these graphs with 10 different random seeds. Finally, we report the average performance and variance across all kinds of experiments.

### 3.2 COMPARISON WITH BASELINES

For node classification, We compare our proposed CTRL with seven baselines: graph coarsening methods (Huang et al., 2021), Random, which randomly selected nodes to form the original graph, core set methods (Herding (Welling, 2009) and K-Center (Sener & Savarese, 2017)), DC-Graph is proposed as a baseline in (Jin et al., 2022b), and the state-of-the-art graph condensation methods GCond (Jin et al., 2022b) and SFGC (Zheng et al., 2023). For graph classification, We compare our proposed CTRL with the above three coreset methods: Random, Herding, and K-Center, the above DC-Graph, and Doscond (Jin et al., 2022a). We report the performances in Table 1 and Table 2.

Based on the results, we have noted the following observations: 1) In most cases of the node classification tasks, CTRL demonstrates better performance than any baseline method. Compared with Gcond, which is also based on gradient matching with significant improvements of up to 6.4%. 2) Across all datasets for graph classification, CTRL still achieves better performance (up to 6.2%) than Doscond, which uses MSE distance as the criterion of gradient matching loss.

Table 1: Performance comparison to baselines in the node classification tasks. *CTRL achieves the highest results in most cases on node classification and lossless results in 3 of 6 datasets.* We report test accuracy (%) on Citeseer, Cora, Ogbn-arxiv, Ogbn-XRT, Flickr and Reddit. **Bold entries** are best results, highlight mark the lossless results.

| Dataset | Ratio | Random | Herding | K-Center | Coarsening | DC-Graph | GCond | SFGC | CTRL(ours) | Whole Dataset |
|---|---|---|---|---|---|---|---|---|---|---|
| Citeseer | 0.90% | $54.4_{\pm4.4}$ | $57.1_{\pm1.5}$ | $52.4_{\pm2.8}$ | $52.2_{\pm0.4}$ | $66.8_{\pm1.5}$ | $70.5_{\pm1.2}$ | $71.4_{\pm0.5}$ | $\mathbf{73.3}_{\pm0.4}$ | $71.7_{\pm0.1}$ |
| | 1.80% | $64.2_{\pm1.7}$ | $66.7_{\pm1.0}$ | $64.3_{\pm1.0}$ | $66.9_{\pm0.9}$ | $59.0_{\pm0.5}$ | $70.6_{\pm0.9}$ | $72.4_{\pm0.4}$ | $\mathbf{73.5}_{\pm0.1}$ | |
| | 3.60% | $69.1_{\pm0.1}$ | $69.0_{\pm0.1}$ | $69.1_{\pm0.1}$ | $65.3_{\pm0.5}$ | $66.3_{\pm1.5}$ | $69.8_{\pm1.4}$ | $70.6_{\pm0.7}$ | $\mathbf{73.4}_{\pm0.2}$ | |
| Cora | 1.30% | $63.6_{\pm3.7}$ | $67.0_{\pm1.3}$ | $64.0_{\pm2.3}$ | $31.2_{\pm0.2}$ | $67.3_{\pm1.9}$ | $79.8_{\pm1.3}$ | $80.1_{\pm0.4}$ | $\mathbf{81.9}_{\pm0.1}$ | $81.2_{\pm0.2}$ |
| | 2.60% | $72.8_{\pm1.1}$ | $73.4_{\pm1.0}$ | $73.2_{\pm1.2}$ | $65.2_{\pm0.6}$ | $67.6_{\pm3.5}$ | $80.1_{\pm0.6}$ | $81.7_{\pm0.5}$ | $\mathbf{81.8}_{\pm0.1}$ | |
| | 5.20% | $76.8_{\pm0.1}$ | $76.8_{\pm0.1}$ | $76.7_{\pm0.1}$ | $70.6_{\pm0.1}$ | $67.7_{\pm2.2}$ | $79.3_{\pm0.3}$ | $81.6_{\pm0.8}$ | $\mathbf{81.8}_{\pm0.1}$ | |
| Ogbn-arxiv | 0.05% | $47.1_{\pm3.9}$ | $52.4_{\pm1.8}$ | $47.2_{\pm3.0}$ | $35.4_{\pm0.3}$ | $58.6_{\pm0.4}$ | $59.2_{\pm1.1}$ | $65.5_{\pm0.7}$ | $\mathbf{65.6}_{\pm0.3}$ | $71.4_{\pm0.1}$ |
| | 0.25% | $57.3_{\pm1.1}$ | $58.6_{\pm1.2}$ | $56.8_{\pm0.8}$ | $43.5_{\pm0.2}$ | $59.9_{\pm0.3}$ | $63.2_{\pm0.3}$ | $66.1_{\pm0.4}$ | $\mathbf{66.5}_{\pm0.3}$ | |
| | 0.50% | $60.0_{\pm0.9}$ | $60.4_{\pm0.8}$ | $60.3_{\pm0.4}$ | $50.4_{\pm0.1}$ | $59.5_{\pm0.3}$ | $64.0_{\pm1.4}$ | $66.8_{\pm0.4}$ | $\mathbf{67.6}_{\pm0.2}$ | |
| Ogbn-XRT | 0.05% | $55.1_{\pm2.1}$ | $60.6_{\pm2.3}$ | $52.7_{\pm3.5}$ | $48.4_{\pm0.4}$ | $60.4_{\pm0.5}$ | $55.1_{\pm0.3}$ | $55.4_{\pm0.1}$ | $\mathbf{61.1}_{\pm0.5}$ | $73.4_{\pm0.1}$ |
| | 0.25% | $64.2_{\pm1.4}$ | $64.4_{\pm1.7}$ | $60.8_{\pm2.2}$ | $51.3_{\pm0.3}$ | $61.5_{\pm0.2}$ | $68.1_{\pm0.4}$ | $67.7_{\pm0.1}$ | $\mathbf{69.4}_{\pm0.4}$ | |
| | 0.50% | $65.7_{\pm2.1}$ | $68.3_{\pm2.2}$ | $62.3_{\pm3.4}$ | $56.9_{\pm0.1}$ | $68.2_{\pm0.3}$ | $69.3_{\pm0.4}$ | $66.5_{\pm0.1}$ | $\mathbf{70.4}_{\pm0.1}$ | |
| Flickr | 0.10% | $41.8_{\pm2.0}$ | $42.5_{\pm1.8}$ | $42.0_{\pm0.7}$ | $41.9_{\pm0.2}$ | $46.3_{\pm0.2}$ | $46.5_{\pm0.4}$ | $46.6_{\pm0.2}$ | $\mathbf{47.1}_{\pm0.1}$ | $47.2_{\pm0.1}$ |
| | 0.50% | $44.0_{\pm0.4}$ | $43.9_{\pm0.9}$ | $43.2_{\pm0.1}$ | $44.5_{\pm0.1}$ | $45.9_{\pm0.1}$ | $47.1_{\pm0.1}$ | $47.0_{\pm0.1}$ | $\mathbf{47.4}_{\pm0.1}$ | |
| | 1.0% | $44.6_{\pm0.2}$ | $44.4_{\pm0.6}$ | $44.1_{\pm0.4}$ | $44.6_{\pm0.1}$ | $45.8_{\pm0.1}$ | $47.1_{\pm0.1}$ | $47.1_{\pm0.1}$ | $\mathbf{47.5}_{\pm0.1}$ | |
| Reddit | 0.01% | $46.1_{\pm4.4}$ | $53.1_{\pm2.5}$ | $46.6_{\pm2.3}$ | $40.9_{\pm0.5}$ | $88.2_{\pm0.2}$ | $88.0_{\pm1.8}$ | $\mathbf{89.7}_{\pm0.2}$ | $89.2_{\pm0.2}$ | $93.9_{\pm0.0}$ |
| | 0.05% | $58.0_{\pm2.2}$ | $62.7_{\pm1.0}$ | $53.0_{\pm3.3}$ | $42.8_{\pm0.8}$ | $89.5_{\pm0.1}$ | $89.6_{\pm0.7}$ | $90.0_{\pm0.3}$ | $\mathbf{90.6}_{\pm0.2}$ | |
| | 0.50% | $66.3_{\pm1.9}$ | $71.0_{\pm1.6}$ | $58.5_{\pm2.1}$ | $47.4_{\pm0.9}$ | $90.5_{\pm1.2}$ | $90.1_{\pm0.5}$ | $90.3_{\pm0.3}$ | $\mathbf{91.9}_{\pm0.4}$ | |

Table 2: The graph classification performance comparison to baselines. *CTRL achieves the highest results in all cases on graph classification and lossless results in 2 of 6 datasets.* We report the ROC-AUC for the first three datasets and accuracies (%) for others. Whole Dataset indicates the performance of the original dataset. **Bold entries** are best results, highlight mark the lossless results.

| Dataset | Ratio | Random | Herding | K-Center | DCG | DosCond | CTRL(ours) | Whole Dataset |
|---|---|---|---|---|---|---|---|---|
| ogbg-molbace | 0.20% | $0.580_{\pm0.067}$ | $0.548_{\pm0.034}$ | $0.548_{\pm0.034}$ | $0.623_{\pm0.046}$ | $0.657_{\pm0.034}$ | $\mathbf{0.716}_{\pm0.025}$ | $0.724_{\pm0.005}$ |
| | 1.70% | $0.598_{\pm0.073}$ | $0.639_{\pm0.039}$ | $0.591_{\pm0.056}$ | $0.655_{\pm0.033}$ | $0.674_{\pm0.035}$ | $\mathbf{0.736}_{\pm0.014}$ | |
| | 8.30% | $0.632_{\pm0.047}$ | $0.683_{\pm0.022}$ | $0.589_{\pm0.025}$ | $0.652_{\pm0.013}$ | $0.688_{\pm0.012}$ | $\mathbf{0.745}_{\pm0.009}$ | |
| ogbg-molbbbp | 0.10% | $0.519_{\pm0.016}$ | $0.546_{\pm0.019}$ | $0.546_{\pm0.019}$ | $0.559_{\pm0.044}$ | $0.581_{\pm0.005}$ | $\mathbf{0.592}_{\pm0.011}$ | $0.646_{\pm0.004}$ |
| | 1.20% | $0.586_{\pm0.040}$ | $0.605_{\pm0.019}$ | $0.530_{\pm0.039}$ | $0.568_{\pm0.032}$ | $0.605_{\pm0.008}$ | $\mathbf{0.629}_{\pm0.006}$ | |
| | 6.10% | $0.606_{\pm0.020}$ | $0.617_{\pm0.003}$ | $0.576_{\pm0.019}$ | $0.579_{\pm0.032}$ | $0.620_{\pm0.007}$ | $\mathbf{0.650}_{\pm0.005}$ | |
| ogbg-molhiv | 0.01% | $0.719_{\pm0.009}$ | $0.721_{\pm0.002}$ | $0.721_{\pm0.002}$ | $0.718_{\pm0.013}$ | $0.726_{\pm0.003}$ | $\mathbf{0.732}_{\pm0.006}$ | $0.757_{\pm0.008}$ |
| | 0.06% | $0.720_{\pm0.011}$ | $0.725_{\pm0.006}$ | $0.713_{\pm0.009}$ | $0.728_{\pm0.002}$ | $0.728_{\pm0.005}$ | $\mathbf{0.734}_{\pm0.008}$ | |
| | 0.30% | $0.721_{\pm0.014}$ | $0.725_{\pm0.003}$ | $0.725_{\pm0.006}$ | $0.726_{\pm0.010}$ | $0.731_{\pm0.004}$ | $\mathbf{0.737}_{\pm0.006}$ | |
| MUTAG | 1.30% | $67.47_{\pm9.74}$ | $70.84_{\pm7.71}$ | $70.84_{\pm7.71}$ | $75.00_{\pm8.16}$ | $82.21_{\pm1.61}$ | $\mathbf{83.06}_{\pm3.15}$ | $88.63_{\pm1.44}$ |
| | 13.30% | $77.89_{\pm7.55}$ | $80.42_{\pm1.89}$ | $81.00_{\pm2.51}$ | $82.66_{\pm0.68}$ | $82.76_{\pm2.31}$ | $\mathbf{83.16}_{\pm3.62}$ | |
| NCI1 | 0.10% | $51.27_{\pm1.22}$ | $53.98_{\pm0.67}$ | $53.98_{\pm0.67}$ | $51.14_{\pm1.08}$ | $56.58_{\pm0.48}$ | $\mathbf{56.69}_{\pm0.69}$ | $71.70_{\pm0.20}$ |
| | 0.60% | $54.33_{\pm3.14}$ | $57.11_{\pm0.56}$ | $53.21_{\pm1.44}$ | $51.86_{\pm0.81}$ | $58.02_{\pm1.05}$ | $\mathbf{58.04}_{\pm1.28}$ | |
| | 3.10% | $58.51_{\pm1.73}$ | $58.94_{\pm0.83}$ | $56.58_{\pm3.08}$ | $52.17_{\pm1.90}$ | $60.07_{\pm1.58}$ | $\mathbf{60.14}_{\pm1.73}$ | |
| CIFAR10 | 0.06% | $15.61_{\pm0.52}$ | $22.38_{\pm0.49}$ | $22.37_{\pm0.50}$ | $21.60_{\pm0.42}$ | $24.70_{\pm0.70}$ | $\mathbf{29.30}_{\pm0.27}$ | $50.75_{\pm0.14}$ |
| | 0.20% | $23.07_{\pm0.76}$ | $28.81_{\pm0.35}$ | $20.93_{\pm0.62}$ | $29.27_{\pm0.77}$ | $30.70_{\pm0.23}$ | $\mathbf{31.21}_{\pm0.20}$ | |
| | 1.10% | $30.56_{\pm0.81}$ | $33.94_{\pm0.37}$ | $24.17_{\pm0.51}$ | $34.47_{\pm0.52}$ | $35.34_{\pm0.14}$ | $\mathbf{35.53}_{\pm0.38}$ | |

This suggests that in gradient matching, aligning either direction or magnitude alone introduces bias, combining both can achieve a better alignment of the gradient trajectories.

### 3.3 ANALYSIS

**Better matching trajectories.** To better illustrate the enhanced capacity of our method in crafting matching trajectories, we conduct experiments on the Reddit dataset with the condensation ratio of 0.5% and quantify the variations between gradients during the graph condensation, yielding the results as depicted in Figure 3(a) and 3(b). More experimental details can be found in Appendix B.3

We can observe that in comparison to GCond, CTRL exhibits two notable improvements: 1) By optimizing both cosine distance and Euclidean distance between gradients, our method effectively addresses the limitations of methods relying solely on matching cosine distance, which fails to align

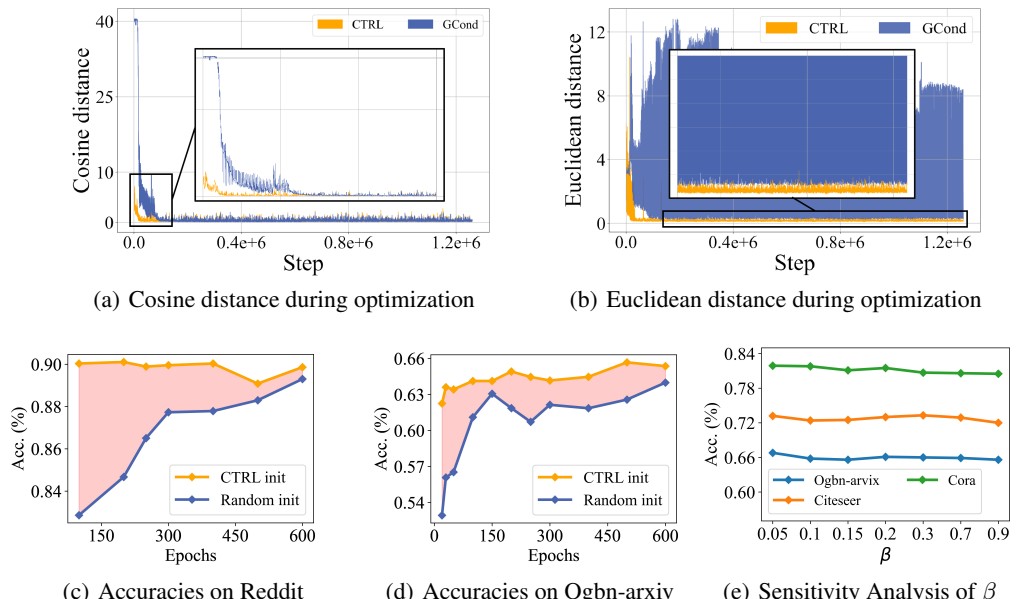

Figure 3: (a) and (b) illustrate the gradient difference generated during the optimization process using GCond and CTRL, step refers to the times of calculating gradient matching losses, demonstrating that our approach not only reduces early-stage cosine distance matching errors but also quickly achieves matching in Euclidean distance. (c) and (d) show the improvement on Ogbn-arxiv and Reddit datasets by employing the initialization method of CTRL.

the gradient magnitude. As shown in Figure 3(b), GCond exhibits minimal capability in optimizing Euclidean distances, with only marginal improvements observed, while CTRL reduces Euclidean distance between gradients to only 3.7% of the original scale, significantly mitigating the variation in gradient magnitudes. 2) Our approach does not introduce side effects in terms of gradient direction alignment. Throughout the entire matching process, both Euclidean distance and cosine distance consistently maintain smaller values. Which can be explained as matching gradient magnitudes aiding in the alignment of gradient directions.

**Quantitative analysis of frequency domain similarity.** We employ multiple metrics to measure how well synthetic graphs generated by CTRL imitate the distribution of frequency domains in the original graph, including the proportion of low-frequency nodes, the mean of high-frequency areas (Tang et al., 2022), spectral peakedness and skewness (Ao et al., 2022), spectral radius (Fan et al., 2022), and the variance of eigenvalues (Sharma et al., 2019). Subsequently, we calculate these metrics separately on the synthetic graph and the original graph, then visually depict the correlation by computing the Pearson coefficient which reflects the overall trend.

We provide the Pearson correlation coefficient and the statistical significance P-value in Table 6. The results of our experiments on multiple datasets show that CTRL-generated synthetic graphs better imitate the spectral distribution in the original graphs compared to GCond, demonstrating that the frequency distribution of the synthetic graph generated by CTRL is more similar to the original graph. This further substantiates the effectiveness of our approach in simulating the frequency domain.

**Ablation study of initialization.** In our investigation of the initialization component of synthetic data, we conduct an ablation study to assess their impact. The results presented in Figure 3(c) and 3(d) are obtained from experiments conducted on the Reddit and Ogbn-arxiv datasets, using a condensation rate of 0.50%. Upon examination of the figures, a clear trend emerges: Initialization of CTRL not only improves performance by approximately 10% at the beginning of training but also leads to significant improvements throughout the distillation process. This suggests that CTRL's initialization method offers a superior starting point for optimizing synthetic data and reduces the additional optimization challenges because of the gradient misalignment in the early stages of gradient matching, ultimately resulting in better consistency throughout the entire gradient matching.

Table 3: Performance across different GNN architectures. Avg. and Std. : the average performance and the standard deviation of the results of APPNP, Cheby, GCN, SAGE and SGC, $\Delta(\%)$ denotes the improvements upon the DG-Graph. We mark the best performance by **bold**.

| Datasets | Methods | Architectures | | | | | | | Statistics | | |
|---|---|---|---|---|---|---|---|---|---|---|---|
| | | MLP | GAT | APPNP | Cheby | GCN | SAGE | SGC | Avg. | Std. | $\Delta(\%)$ |
| Cora | DC-Graph | 67.2 | - | 67.1 | 67.7 | 67.9 | 66.2 | 72.8 | 68.3 | 2.6 | - |
| | GCond | 73.1 | 66.2 | 78.5 | 76.0 | 80.1 | 78.2 | 79.3 | 78.4 | 4.9 | ↑10.1 |
| | CTRL | 77.4 | 66.7 | 80.8 | 77.0 | 81.6 | 79.0 | 81.5 | **80.0** | **2.0** | ↑**11.7** |
| Ogbn-arxiv | DC-Graph | 59.9 | - | 60.0 | 55.7 | 59.8 | 60.0 | 60.4 | 59.2 | **2.0** | - |
| | GCond | 62.2 | 60.0 | 63.4 | 54.9 | 63.2 | 62.6 | 63.7 | 61.6 | 3.7 | ↑2.4 |
| | CTRL | 64.1 | 60.3 | 65.3 | 57.7 | 65.8 | 62.9 | 66.1 | **63.6** | 3.5 | ↑**4.4** |
| Flickr | DC-Graph | 43.1 | - | 45.7 | 43.8 | 45.9 | 45.8 | 45.6 | 45.4 | **0.9** | - |
| | GCond | 44.8 | 40.1 | 45.9 | 42.8 | 47.1 | 46.2 | 46.1 | 45.6 | 1.6 | ↑0.2 |
| | CTRL | 42.6 | 41.0 | 46.2 | 43.1 | 47.2 | 46.5 | 46.7 | **45.9** | 1.6 | ↑**0.5** |
| Reddit | DC-Graph | 50.3 | - | 81.2 | 77.5 | 89.5 | 89.7 | 90.5 | 85.7 | **5.9** | - |
| | GCond | 42.5 | 60.2 | 87.8 | 75.5 | 89.4 | 89.1 | 89.6 | 86.3 | 6.1 | ↑0.6 |
| | CTRL | 43.2 | 60.0 | 87.9 | 75.3 | 90.3 | 89.1 | 89.7 | **86.5** | 6.3 | ↑**0.8** |

Table 4: Comparison of the cross-architecture generalization performance between GCond (on the left of /) and CTRL (on the right of /) on Cora(a) and Ogbn-arxiv(b). **Bold** entries are the best results. ↑/↓ : our method show increase or decrease performance. Overall, graphs condensed with CTRL by different GNNs exhibit stronger transfer performance on other architectures than GCond.

(a) Cora, ratio=2.60%

| C/T | APPNP | Cheby | GCN | SAGE | SGC |
|---|---|---|---|---|---|
| APPNP | 72.1/**76.1**↑ | 60.8/**75.1**↑ | 73.5/**76.7**↑ | 72.3/**72.7**↑ | 73.1/**78.0**↑ |
| Cheby | 75.3/**78.5**↑ | 71.8/**75.8**↑ | **76.8**/74.1↓ | **76.4**/75.2↓ | 75.5/**75.6**↑ |
| GCN | 69.8/**72.5**↑ | 53.2/**62.4**↑ | 70.6/**72.3**↑ | 60.2/**60.6**↑ | 68.7/**73.1**↑ |
| SAGE | 77.1/**77.2**↑ | 69.3/**75.9**↑ | 77.0/**79.3**↑ | **76.1**/75.7↓ | 77.7/**79.1**↑ |
| SGC | 78.5/**80.9**↑ | 76.0/**77.5**↑ | 80.1/**80.7**↑ | **78.2**/74.8↓ | **79.3**/76.5↓ |

(b) Ogbn-arxiv, ratio=0.05%

| C/T | APPNP | Cheby | GCN | SAGE | SGC |
|---|---|---|---|---|---|
| APPNP | 60.3/**62.5**↑ | 51.8/**57.3**↑ | 59.9/**63.6**↑ | 59.0/**61.3**↑ | 61.2/**62.5**↑ |
| Cheby | 57.4/**59.2** ↑ | 53.5/**55.2**↑ | 57.4/**59.0**↑ | **57.1**/55.2↓ | **58.2**/57.7↓ |
| GCN | 59.3/**61.4**↑ | 51.8/**55.8**↑ | 60.3/**61.1**↑ | 60.2/**60.3**↑ | 59.2/**61.9**↑ |
| SAGE | 57.6/**60.7**↑ | **53.9**/53.3↓ | 58.1/**62.9**↑ | **57.8**/55.5↓ | 59.0/**61.7**↑ |
| SGC | 57.6/**60.6**↑ | 53.9/**55.9**↑ | 58.1/**61.8**↑ | 57.8/**58.9**↑ | 59.0/**61.6**↑ |

**Sensitivity analysis of $\beta$.** To assess the sensitivity of $\beta$, we conduct a series of experiments on the Cora, Citeseer, and Ogbn-arxiv datasets, with distillation ratios of 1.3%, 0.9%, and 0.25%, respectively. The experimental results indicate that our method is not sensitive to the $\beta$, as shown in Figure 3(e). Specifically, when the value of $\beta$ varies, the change in accuracy on the test set for the generated synthetic data does not exceed 1.5%. This validates that CTRL itself possesses good robustness and does not overly rely on the choice of the $\beta$.

**Cross-architecture generalization analysis.** To further demonstrate the generalization ability of the graph compression process, we employ a GNN model to condense a graph and train various GNN architectures on it, subsequently evaluating their performance to analyze the transferability of the learned representations. Specifically, we selected APPNP, GCN, SGC, GraphSAGE (Hamilton et al., 2017), Cheby and GAT (Veličković et al., 2017), as well as a standard MLP. As reported in Table 3, compared to GCond, the improvement of CTRL is up to 2%, demonstrating that the finer-grained gradient matching does not lead condensed graph to overfit in a single neural network architecture.

**Versatility of CTRL.** We also evaluate our method on Cora and Ogbn-arxiv graph datasets using five different graph neural network architectures: APPNP, ChebyNet, GCN, GraphSAGE, and SGC, as done in Jin et al. (2022b). To be more specific, we first condense each graph dataset separately using each architecture, then evaluate the condensed graphs on all five architectures and report their performances in Table 4. The results show that our approach achieves improved performance over GCond on the majority of model-dataset combinations, demonstrating that CTRL is versatile and achieves consistent gains across diverse architectures and datasets.

**Neural architecture search evaluation.** We extend our investigation to neural architecture search (NAS) to comprehensively assess the performance of our proposed CTRL. Following the setting

Table 5: Neural Architecture Search. Methods are compared in validation accuracy correlation and test accuracy obtained by searched architecture. Whole means the architecture is searched using whole dataset. CTRL achieved better performance than GCond in all 3 datasets.

Table 6: Comparing the distribution of high-frequency and low-frequency signals in synthetic graph generated by CTRL and GCond with the original graph.

| Dataset | Pearson Correlation / Performance(%) | | | | Whole |
|---|---|---|---|---|---|
| | Random | Herding | GCond | CTRL | Per.(%) |
| Cora | 0.40/82.9 | 0.21/82.9 | 0.76/83.1 | **0.86/83.2** | 82.6 |
| Citeseer | 0.56/71.4 | 0.29/71.3 | 0.79/71.3 | **0.93/72.0** | 71.6 |
| Pubmed | 0.55/80.0 | 0.21/79.9 | 0.81/79.7 | **0.83/80.1** | 80.5 |

| Dataset | Pearson Correlation / P-value | |
|---|---|---|
| | GCond | CTRL |
| Cora | 0.90/0.016 | **0.96/0.003** |
| Citeseer | 0.92/0.009 | **0.95/0.003** |
| Pumbed | 0.89/0.016 | **0.90/0.014** |

in Jin et al. (2022b), we search 480 architectures on condensed graphs of the Cora, Citeseer, and Pubmed datasets. The results reported in Table 5 demonstrate the consistent superiority of our method, evidenced by higher Pearson correlation coefficients (Li et al., 2022a) and improved test performance, with enhancements of up to 0.14 and 0.4%, respectively. These results underscore the architectures searched by CTRL are more efficient when trained on the full graph datasets.

## 4 RELATED WORK

**Dataset Distillation & Dataset Condensation.** Dataset distillation (DD) and dataset condensation (DC) are two techniques to reduce the size and complexity of large-scale datasets for training deep neural networks. DD condenses a large-scale dataset into a small synthetic one such that the model trained on the latter achieves comparable performance (Wang et al., 2018; Bohdal et al., 2020; Nguyen et al., 2020). DC aims to improve the efficiency of DD, a common approach is matching the gradients (Zhao et al., 2020; Zhao & Bilen, 2021). Both techniques have been applied to various data modalities, such as images, text, speech, and graphs (Zhang et al., 2023), and have shown promising results in reducing the training cost (Liu et al., 2022; Wu et al., 2023b). However, there are also many challenges and open questions in DC, such as how to design an effective matching strategy, (Sucholutsky & Schonlau, 2020). In this work, we focus on designing a novel matching strategy for more efficient graph dataset condensation.

**Graph Coarsening & Graph Sparsification.** Graph coarsening minimizes the size of graph data while preserving its basic properties (Chen et al., 2022a; Kammer & Meintrup, 2022; Kelley & Rajamanickam, 2022) by grouping similar nodes into supernodes. Graph Sparsification reduces the number of edges to make the graph sparser (Li et al., 2022b; Yu et al., 2022a; Chen et al., 2022b), as there are many redundant relations in the graphs. Both two methods are based on the idea of coreset selection (Chen et al., 2012; Campbell & Broderick, 2019), aiming to remove less important information from the original graph. However, these methods rely heavily on heuristic unsupervised techniques, resulting in poor generalization performance in downstream tasks. Moreover, they are unable to condense graphs with extremely low condensation ratios.

## 5 CONCLUSION

We introduce CTRL, a novel approach for graph condensation through finer-grained gradient matching and more rational initialization. This approach exhibits robustness across various scenarios and achieves lossless results on Cora, Citeseer, Flickr, ogbn-molbace and ogbn-molbbbp. More importantly, our method is able to be effectively integrated into all gradient-based dataset condensation methods easily, promising to advance future research in this field. This paper provides comprehensive experimental analysis, contributing valuable insights for the broader research community.

**Limitations and future work.** Our current method, while effective for gradient matching, may cause the potential loss of informative properties, like heterogeneity, during the condensation process. This limitation can adversely affect performance in subsequent tasks. To overcome these constraints, we plan to explore a more versatile method in our future research, aiming to preserve informative properties during the condensation process, ultimately enhancing performance in downstream tasks.

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

# A  DATASETS AND IMPLEMENTATION DETAILS

## A.1  DATASETS

We evaluate CTRL on three transductive datasets: Cora, Citeseer (Kipf & Welling, 2016), Ogbn-arxiv (Hu et al., 2020), Ogbn-xrt (Chien et al., 2021), and two inductive datasets: Flickr (Zeng et al., 2020), and Reddit (Hamilton et al., 2017). We obtain all datasets from PyTorch Geometric(Fey & Lenssen, 2019) with publicly available splits and consistently utilize these splits in all experiments. For graph classification, we use multiple molecular datasets from Open Graph Benchmark (OGB) (Hu et al., 2020) and TU Datasets (MUTAG and NCI1) (Morris et al., 2020) for graph-level property classification, and one superpixel dataset CIFAR10 (Dwivedi et al., 2020). In addition, we use Pubmed(Kipf & Welling, 2016) in our neural architecture search (NAS) experiments. Dataset statistics are shown in Table 7 and 8.

Table 7: Dataset statistics(node classification). The first four are transductive datasets and the last two are inductive datasets.

| Dataset | #Nodes | #Edges | #Classes | #Features | Training/Validation/Test |
|---|---|---|---|---|---|
| Cora | 2,708 | 5,429 | 7 | 1,433 | 140/500/1000 |
| Citeseer | 3,327 | 4,732 | 6 | 3,703 | 120/500/1000 |
| Ogbn-arxiv | 169,343 | 1,166,243 | 40 | 128 | 90,941/29,799/48,603 |
| Ogbn-xrt | 169,343 | 1,166,243 | 40 | 768 | 90,941/29,799/48,603 |
| Flickr | 89,250 | 899,756 | 7 | 500 | 44,625/22312/22313 |
| Reddit | 232,965 | 57,307,946 | 210 | 602 | 15,3932/23,699/55,334 |

Table 8: Dataset statistics(graph classification).

| Dataset | Type | #Clases | #Graphs | #Avg.Nodes | #Avg. Edges |
|---|---|---|---|---|---|
| CIFAR10 | Superpixel | 10 | 60,000 | 117.6 | 941.07 |
| ogbg-molhiv | Molecule | 2 | 41,127 | 25.5 | 54.9 |
| ogbg-molbace | Molecule | 2 | 1,513 | 34.1 | 36.9 |
| ogbg-molbbbp | Molecule | 2 | 2,039 | 24.1 | 26.0 |
| MUTAG | Molecule | 2 | 188 | 17.93 | 19.79 |
| NCI1 | Molecule | 2 | 4,110 | 29.87 | 32.30 |

## A.2  IMPLEMENTATION DETAILS

During the implementation of CTRL, considering datasets with a large number of nodes, where the feature distribution plays a crucial role in the overall data, such as Ogbn-arxiv, Ogbn-xrt, and Reddit, we employ the specific initialization method depicted in Sec. 2.2 on them. Additionally, due to the substantial volume of data and numerous gradient matching iterations, we introduce a threshold $tau$ during the matching process. We match gradients smaller than this threshold on both direction and magnitude, while gradients exceeding the threshold were matched solely based on their directions. Through extensive experimentation, we observe that this strategy reduced computational costs and led to further performance improvements. For other datasets, due to the smaller dataset size, we utilize both gradient magnitude and direction for matching throughout the entire matching process and we directly employ random sampling for initialization. For graph classification tasks, we employ gradient magnitude and direction matching throughout the graph condensation and adopt a strategy of random sampling for initialization. And the beta in Figure 3(c), 3(d) are 0.1 and 0.15, respectively.

## A.3  HYPER-PARAMETER SETTING

For node classification, without specific mention, we adopt a 2-layer SGC (Wu et al., 2019) with 256 hidden units as the GNN used for gradient matching. We employ a multi-layer perceptron (MLP) as

the function $g_\Phi$ models the relationship between $\mathbf{A}'$ and $\mathbf{X}'$. Specifically, we adopt a 3-layer MLP with 128 hidden units for small graphs (Cora and Citeseer) and 256 hidden units for large graphs (Flickr, Reddit, and Ogbn-arxiv). We tune the training epoch for CTRL in a range of 400, 600, 1000. For the choices of condensation ratio $r$, we divide the discussion into two parts. The first part is about transductive datasets. For Cora and Citeseer, since their labeling rates are very small (5.2% and 3.6%, respectively), we choose r to be 25%, 50%, 100% of the labeling rate. Thus, we finally choose 1.3%, 2.6%, 5.2% for Cora and 0.9%, 1.8%, 3.6% for Citeseer. For Ogbn-arxiv and Ogbn-xrt, we choose r to be 0.1%, 0.5%, 1% of its labeling rate (53%), thus being 0.05%, 0.2%, 0.5%. The second part is about inductive datasets. As the nodes in the training graphs are all labeled in inductive datasets, we choose 0.1%, 0.5%, 0.1% for Flickr and 0.05%, 0.1%, 0.2% for Reddit. The $beta$ that measures the weight of the direction and magnitude is in 0.1, 0.2, 0.5, 0.7, 0.9, please refer to the experiments Sec. 3.3 for a detailed sensitivity analysis. For graph classification, we vary the number of learned synthetic graphs per class in the range of 1, 10, 50 (1, 10 for MUTAG, in the case where the condensation ratio has already reached 13.3% with a factor of 10, and we will refrain from further increasing the condensation ratio to ensure the effectiveness of data condensation.) and train a GCN on these graphs. The $beta$ that measures the weight of the direction and magnitude is between 0.05 and 0.9 for node classification tasks, while between 0.1 and 0.9 for graph classification tasks.

# B   MORE DETAILS ABOUT QUANTITATIVE ANALYSIS

## B.1   SELECTION OF THE THRESHOLD.

We determine the optimal frequency domain threshold by first calculating the spectral gap between the maximum and minimum eigenvalues of the graph Laplacian matrix. Then search over threshold values within a range centered around the median eigenvalue spanning a fraction of the spectral gap. For each potential threshold, we compute the graph signal reconstruction error by splitting the signal into high and low-frequency components based on the threshold, recombining the components, and quantifying the Euclidean distance between the original and reconstructed signal vectors (Ortega et al., 2018; Ramírez et al., 2021; Leus et al., 2023). The optimal threshold corresponds to the one minimizing this reconstruction error across all tested values. This optimal data-driven threshold provides the split that best preserves the graph signal when separating it into high and low-frequency components, as indicated by the ability to reconstruct the signal with minimal distortion.

## B.2   MULTIPLE METRIC.

We select six complementary metrics to provide a comprehensive evaluation of the distribution of high and low-frequency signals between the synthetic and original graphs. The proportion of low-frequency nodes directly quantifies the relative abundance of high versus low-frequency signals. Meanwhile, the mean of the high-frequency signal region for each feature dimension can describe the right-shift phenomenon in the whole spectrum (Tang et al., 2022) and get a measure of the overall strength of the high-frequency content. Additionally, spectral peakedness and skewness directly characterize the shape of the frequency distribution (Gallier, 2016; Ao et al., 2022). Spectral radius indicates the overall prevalence of high-frequency signals, with a larger radius corresponding to more high-frequency content (Gao & Hou, 2017; Fan et al., 2022). Finally, the variance of eigenvalues measures the spread of the frequency distribution (Min & Chen, 2016; Sharma et al., 2019).

Together, these metrics enable multi-faceted quantitative analysis of the frequency distribution from diverse perspectives, including intuitive reflection of frequency distribution, spectral characterization to assess distribution shapes, and statistical distribution properties like spread and central tendency. To provide a more intuitive summarized measure of the similarity between the synthetic and original graphs across the six metrics, we first normalize all these metrics by simply multiplying or dividing by powers of 10 in order to mitigate the impact of scale differences, then we compute the Pearson correlation coefficient which reflects the overall trend, complementing the individual metric comparisons.

## B.3   MORE DETAILS ABOUT MATCHING TRAJECTORIES.

In the training process, the optimization loss for synthetic data is obtained by training the GNN model on subgraphs composed of nodes from each class. To better evaluate the optimization effectiveness,

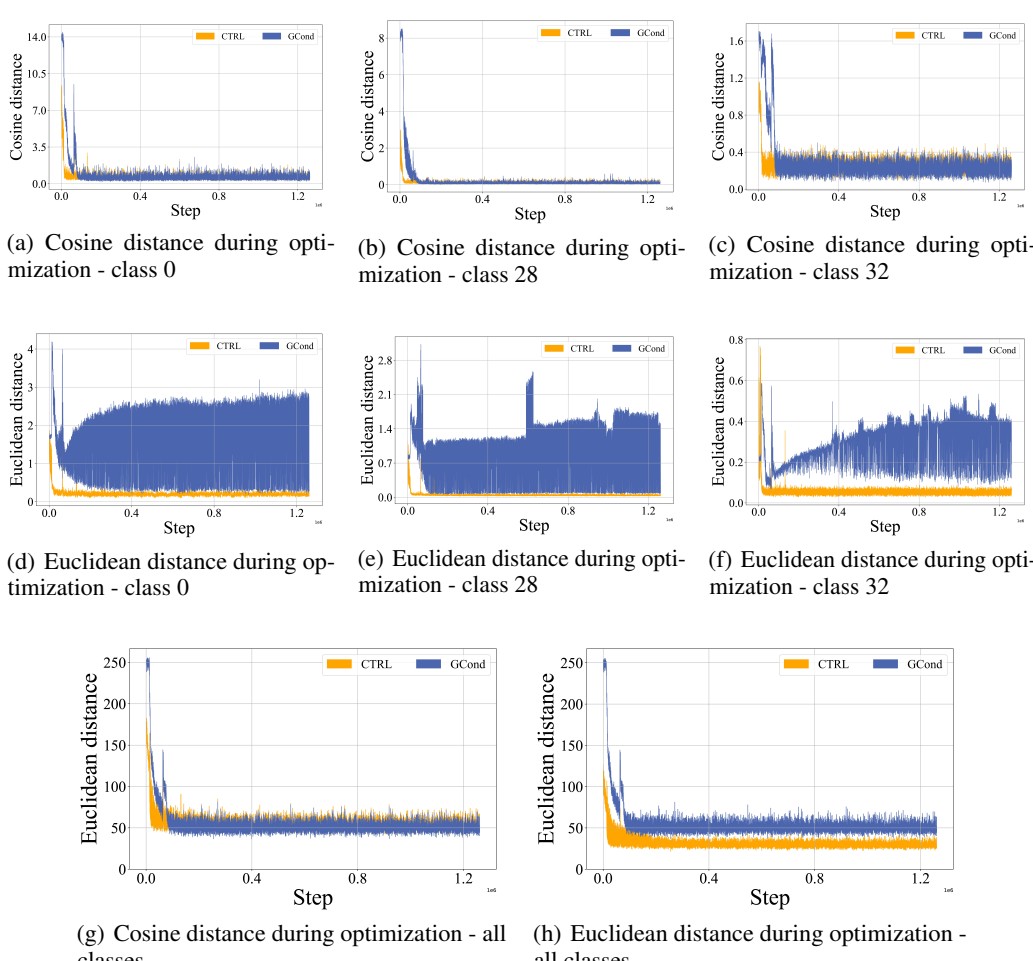

Figure 4: (a)(b)(c)(d)(e)(f)(g) and (h) illustrate the gradient differences generated during the optimization process using GCond and CTRL. The first six images depict the gradient disparities produced by nodes from various classes. Subsequently, the last two images (g) and (h), showcase the overall gradient differences weighted according to the GCond method. Notably, these results align closely with the conclusions drawn from previous experiments Sec. 3.3.

we record the gradient differences generated during training between nodes of each class in both the synthetic and original data. In previous experiments Sec. 3.3, due to space limitations (Reddit has a total of 40 classes), we presented results for class 3 with a relatively high proportion of nodes. In this context, to make the experimental results more representative, as depicted in Figure 4, we showcase representative results for classes with a lower proportion of nodes (class 32), classes with a moderate proportion of nodes (class 28), and another class with a higher proportion of nodes (class 28). We also present corresponding experimental results for the method of computing the total loss by weighting the losses generated by different-class nodes according to GCond.

## B.4 DETAILS ABOUT GRADIENT MAGNITUDES AND GRAPH SPECTRAL DISTRIBUTION.

Employing the methods outlined in Tang et al. (2022) to create artificial graphs, we first initialize the composite graphs with three graph models, Erdos-Renyi (Erdős et al., 2013), Barabasi-Albert (Stamatelatos & Efraimidis, 2022), and Watts-Strogatz (Song & Wang, 2014). Subsequently, we conducted 3*4 sets of experiments on four commonly used GNN models: SGC (Wu et al., 2019), ChebNet (Defferrard et al., 2016), GCN, and APPNP (Gasteiger et al., 2018), and repeat each experiment 1000 times by varying the standard deviation of the Gaussian distribution and the random seed. The results reveal a fairly strong correlation between the high-frequency area and the

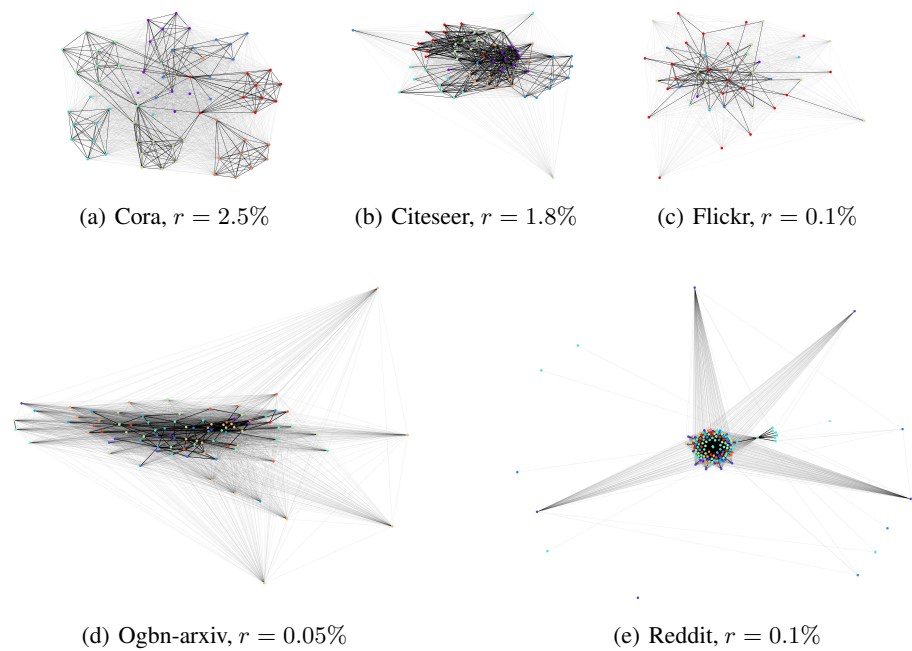

(a) Cora, $r = 2.5\%$      (b) Citeseer, $r = 1.8\%$      (c) Flickr, $r = 0.1\%$

(d) Ogbn-arxiv, $r = 0.05\%$      (e) Reddit, $r = 0.1\%$

Figure 5: When the size of the original datasets is relatively small(a)(b), the resulting composite dataset tends to exhibit fewer outliers. In contrast, when working with compressed results of large datasets(c)(d)(e), it is more likely to encounter a higher number of outliers

gradient magnitudes, as shown in Figure 2, with the high-frequency area increasing gradually, the gradient size generated has an obvious upward trend. Furthermore, 8 of the 12 experiments showed a correlation greater than 0.95. We conducted several experiments in this study, utilizing the following implementation details:

We initialize a list of 1,000 random seeds ranging from 0 to 100,000,000. With a fixed random seed of 0, we created random synthetic graphs using the Erdos-Renyi (Erdős et al., 2013) (edge probability 0.2), Barabasi-Albert (2 edges per new node) (Stamatelatos & Efraimidis, 2022), and Watts-Strogatz (Song & Wang, 2014) (4 initial neighbors per node, rewiring probability 0.2) models. For each random seed, we followed the same procedure: We selected a bias term from [1, 2, 3, 4, 5] in sequence and then used these different bias terms to generate node features following a Gaussian distribution. With a fixed 200 nodes and 128 feature dimensions, we computed the mean of high-frequency area for each feature dimension based on the node feature matrix and adjacency matrix of the synthetic graph. We then trained a simple 2-layer SGC (Wu et al., 2019), ChebNet (Defferrard et al., 2016), GCN, and APPNP (Gasteiger et al., 2018) for 50 epochs, recording the average gradient magnitude. We calculated the Spearman correlation coefficient using the 1,000 rounds of results obtained. The results presented in Figure 2 were from two rounds under the Erdos-Renyi model, while Table 9 shows the corresponding results under different graph models and GNNs.

Table 9: Spearman correlation coefficients of high-frequency area and gradient amplitude under different graph models and GNN architectures. Strong correlations were shown under almost all combinations

|  | SGC | Cheby | APPNP | GCN |
|---|---|---|---|---|
| Erdos-Renyi | 0.953 | 0.965 | 0.955 | 0.859 |
| Barabasi-Albert | 0.955 | 0.961 | 0.875 | 0.818 |
| Watts-Strogatz | 0.946 | 0.955 | 0.939 | 0.679 |

# C  MORE EXPERIMENTS

## C.1  VISUALIZATIONS

To better understand the effectiveness of CTRL, we can visualize the condensed graphs in node classification tasks. It is worth noting that while utilizing the CTRL technique to generate synthetic graphs, it is possible to encounter outliers in contrast with GCOND, especially when working with extensive datasets. This phenomenon arises due to the difficulty in learning and filtering out high-frequency signals present in larger datasets, given the low-pass filtering nature of GNNs. The presence of outliers in the condensed graphs generated by the CTRL method implies that some of the high-frequency signals in the original data are well-preserved in the synthesized graph (Sandryhaila & Moura, 2013a). Previous research has demonstrated the importance of both high and low-frequency signals in graph data for effective GNN training (Bo et al., 2021; Wu et al., 2023a). This further substantiates the validity of this observation.

## C.2  TIME COMPLEXITY AND RUNTIME

**Time complexity**. For simplicity, let the number of MLP layers in the adjacency matrix generator be $L$, and all the hidden units are $d$. In the forward process, we first calculate the $A'$, which has the complexity of $O(N'^2 d^2)$. Second, the forward process of GNN on the original graph has a complexity of $O(mLNd^2)$, where m denotes the sampled size per node in training. Third, the complexity of training on the condensed graph is $O(LN'd)$. Then, taking into account additional matching metrics calculations, the complexity of the gradient matching strategy is $O(2|\theta| + |A'| + |X'|)$.

The overall complexity of CTRL can be represented as $O(N'^2 d^2) + O(mLNd^2) + O(LN'd) + O(2|\theta| + |A'| + |X'|)$. Note $N' < N$, we can drop the terms that only involve $N'$ and constants (e.g., the number of $L$ and $m$). The final complexity can be simplified as $O(mLNd^2)$, thus it can be seen that although the matching process is much finer, the complexity of CTRL still is linear to the number of nodes in the original graph.

**Running time**. We report the running time of the CTRL in two kinds of six groups of classification tasks. For node classification tasks, we vary the condensing ratio r in the range of $\{0.9\%, 1.8\%, 3.6\%\}$ for Citeseer, $\{1.3\%, 2.6\%, 5.2\%\}$ for Cora, $\{0.1\%, 0.5\%, 1.0\%\}$ for Ogbn-arxiv, all experiments are conducted five times on one single V100-SXM2 GPU. For graph classification tasks, We vary the condensing ratio *r* in the range of $\{0.2\%, 1.7\%, 8.3\%\}$ for Ogbg-molbace, $\{0.1\%, 1.2\%, 6.1\%\}$ for Ogbg-molbbbp, $\{0.1\%, 0.5\%, 1.0\%\}$ for Ogbg-molhiv, all experiments are repeated five times on one single A100-SXM4 GPU.

We also conduct experiments with GCond and Docscond using the same settings, respectively. As shown in Table 10 and 11, our approach achieved a similar running time to GCond and Doscond, note that the comparison procedure here improves the cosine distance calculation of GCond, otherwise, our method would be faster on small datasets.

Table 10: Runing time on Citeseer, Cora and Ogbn-arxiv for 50 epochs.

| Dataset | $r$ | GCond | CTRL |
|---------|-----|-------|------|
| Citeseer | 0.9% | 68.7±2.4s | 71.3±2.5s |
| | 1.8% | 70.2±2.5s | 73.7±2.6s |
| | 3.6% | 78.1±2.3s | 88.6±2.1s |
| Cora | 1.3% | 76.4±3.3s | 76.8±2.8s |
| | 2.6% | 77.7±2.2s | 78.7±3.4s |
| | 5.2% | 85.3±3.4s | 89.2±2.5s |
| Ogbn-arxiv | 0.1% | 939.3±4.9s | 967.6±5.8s |
| | 0.5% | 1008.4±3.1s | 1033.4±4.2s |
| | 1.0% | 1061.3±2.9s | 1087.6±1.8s |

Table 11: Runing time on Ogbg-molbace, molbbbp and molhiv for 100 epochs.

| Dataset | $r$ | Doscond | CTRL |
|---------|-----|---------|------|
| Ogbg-molbace | 0.2% | 18.1±1.4s | 21.3±1.5s |
| | 1.7% | 23.1±1.5s | 27.4±1.6s |
| | 8.3% | 26.6±1.3s | 29.5±1.1s |
| Ogbg-molbbbp | 0.1% | 17.2±1.3s | 18.9±1.8s |
| | 1.2% | 20.2±1.2s | 23.4±1.4s |
| | 6.1% | 22.1±1.4s | 24.9±1.5s |
| Ogbg-molhiv | 0.1% | 39.6±1.9s | 42.3±1.8s |
| | 0.5% | 40.2±1.1s | 43.1±2.2s |
| | 1.0% | 40.8±1.9s | 43.9±1.8s |

## D  MORE RELATED WORK

**Graph signal processing.** Our work is also related to graph signal processing, which studies the analysis and processing of signals defined on graphs (Ortega et al., 2018; Nica, 2018). Graph signal processing extends classical signal processing concepts like frequency, filtering, and sampling to graph signals (Shuman et al., 2013; Sandryhaila & Moura, 2013b), which are functions that assign values to the nodes or edges of a graph, providing tools for feature extraction, denoising, compression, and learning on graph-structured data (Hammond et al., 2009; Loukas et al., 2015).

**Graph neural networks (GNNs).** As the generalization of deep neural networks to graph data, Graph Neural Networks (GNNs) enhance the representation of individual nodes by utilizing information from their neighboring nodes(Kipf & Welling, 2016; You et al., 2019; Wu et al., 2022). Due to their powerful capability in handling graph-structured data, GNNs have achieved remarkable performance on various real-world tasks, such as social networks(Hoff et al., 2002; Perozzi et al., 2014), physical(Bear et al., 2020; Raffo et al., 2021; McKay et al., 2022), and chemical interactions(He et al., 2016; Battaglia et al., 2018; Wu et al., 2020; Zhou et al., 2020), and knowledge graphs(Noy et al., 2019; Hogan et al., 2021; Ji et al., 2021).

