# OpenReview forum: "CTRL: Graph condensation via crafting rational trajectory matching"
_ICLR.cc/2024/Conference — Submitted to ICLR 2024_

### Official Review · Reviewer_kSU8 · 2023-10-28

**Soundness:** 3 good
**Presentation:** 3 good
**Contribution:** 1 poor
**Rating:** 3
**Confidence:** 4

**Summary:**

This paper extends the gradient-matching approach for graph condensation by not only leveraging cosine similarity to align gradient directions but also incorporating a regularization term based on Euclidean distance. Empirical evidence suggests that this added Euclidean distance component aids in aligning the frequency distribution with that of the original data. Experimental outcomes demonstrate that the introduced method marginally outperforms existing benchmarks in both node and graph classification evaluations.

**Strengths:**

S1: The exploration of integrating the Euclidean distance component into gradient matching, and its potential impact on frequency alignment, offers insights that could be valuable to the community.

S2: The paper is well-organized and presents a clear narrative. The experiments showcased are comprehensive and thoughtfully executed.

**Weaknesses:**

W1: The core techniques introduced in this paper, including the Euclidean distance component and replacing the random initialization of the synthetic data by selecting nodes from sub-clusters formulated by K-means, have all been previously adopted for dataset condensation [1][2]. This makes the proposed method appear to be a straightforward mash-up of multiple existing methods in dataset condensation, resulting in limited novelty.

W2: Although the authors repeatedly emphasize that their method considers the impact of matching gradient magnitude rather than only matching directions, in actual implementation, they introduced a regularization term based on the Euclidean distance between gradients. This is somewhat inconsistent with their statement and might be misleading, as the Euclidean distance also considers differences beyond just the magnitude of vectors.

W3: From the presented results, it seems that the proposed method does not offer significant improvements over the compared state-of-the-art.

[1] Wei Jin, Xianfeng Tang, Haoming Jiang, Zheng Li, Danqing Zhang, Jiliang Tang, Bing Yin: Condensing Graphs via One-Step Gradient Matching. KDD 2022: 720-730

[2] Yanqing Liu, Jianyang Gu, Kai Wang, Zheng Zhu, Wei Jiang, Yang You: DREAM: Efficient Dataset Distillation by Representative Matching. CoRR abs/2302.14416 (2023)

**Questions:**

Q1: Following up on W2, in Equation 3, I'm curious about why you did not directly address the magnitude differences between gradients. That is,  instead of using the proposed Euclidean distance between gradients, $ \lVert \mathbf{G}{\mathbf{i}}^{\mathcal{S}}-\mathbf{G}{\mathbf{i}}^{\mathcal{T}} \rVert $, why not directly employ $ \lVert \mathbf{G}{\mathbf{i}}^{\mathcal{S}} \rVert  - \lVert\mathbf{G}{\mathbf{i}}^{\mathcal{T}}\lVert $?

---

> ### Author Response · Authors · 2023-11-14
> **Response to Reviewer kSU8(1/2)**
>
> Thank you for your recognition of our work in revealing the relationship between gradient magnitude and frequency domain, as well as our experiments. We sincerely thank you for the detailed comments and insightful questions. We will respond to your comments as follows.
>
> **Weaknesses 1: The proposed method appears to be a straightforward mash-up of multiple existing methods in dataset condensation, resulting in limited novelty.**
>
> **A1:**
>
> Thank you very much for your comments on our method.
>
> In summary, our approach differs significantly from previous methods, not merely as a straightforward mash-up. While there might be some formal similarities, our methodology is grounded in **entirely different principles and analytical processes**. Furthermore, we provide robust experimental evidence to support our claims.
>
> First of all, DosCond [A] does not provide a detailed explanation for the choice of using Euclidean distance. In practice, although the paper employs Euclidean distance, it is worth noting that the official code on GitHub utilizes MSE distance for experimentation. In contrast, our decision to use Euclidean distance **stems from** pre-experiments demonstrating **the strong correlation** between gradient magnitude measured with Euclidean distance and the frequency domain distribution of the graph. Furthermore, experimental results confirm that our approach better preserves the frequency domain distribution of the graph.
>
> In addition, the current initialization methods within the realm of graph compression exhibit several issues. For instance, both real data sampling and core-set-based initialization pose challenges. The former is overly simplistic and may result in **uneven sampling**, while the latter necessitates training the GNN on the **entire dataset** to obtain improved embeddings for core-set selection, which is particularly challenging for large datasets. In contrast, our method only requires clustering of data within each class, significantly reducing the computational resource requirements. Experimental results also demonstrate that our approach effectively captures the feature distribution of the original data, offering an enhanced starting point for synthetic graph optimization with almost no additional computational cost.
>
> To further illustrate the distinctions between our approach and previous methods, we conducted experiments by directly integrating our method with existing approaches (improving GCOND by the sampling method in Dream[B], and altering the gradient matching method based on MSE distance, respectively). The results indicate that this direct combination led to counterproductive outcomes. In contrast, our paper shows that both components of CTRL can effectively improve the performance of existing methods.
>
> Note that all results in the table are averaged over three repetitions for accuracy and consistency.
>
> |                            | GCond | GCond+Dream           |       GCond(mse)              |
> | - | - | - | - |
> | Cora (ratio = 1.30%)       | 79.9  | 78.8±1.1  ${\downarrow}$1.1 | 78.4±0.8  ${\downarrow}$1.3 |
> | Cora (ratio = 2.60%)       | 80.0  | 78.0±0.3  ${\downarrow}$2.1 | 72.4±0.5  ${\downarrow}$7.6 |
> | Cora (ratio = 5.20%)       | 79.1  | 77.8±0.2  ${\downarrow}$1.2 | 74.2±0.1  ${\downarrow}$4.9 |
> | Citeseer (ratio = 0.90%)   | 70.7  | 68.3±0.3  ${\downarrow}$2.4 | 64.8±0.5  ${\downarrow}$5.9 |
> | Citeseer (ratio = 1.80%)   | 70.9  | 68.1±1.1  ${\downarrow}$2.8 | 62.3±1.0  ${\downarrow}$8.6 |
> | Citeseer (ratio = 3.60%)   | 70.3  | 68.0±1.1  ${\downarrow}$2.3 | 63.9±0.8  ${\downarrow}$6.4 |
> | Flickr (ratio = 0.10%)     | 46.5  | 43.6±0.2  ${\downarrow}$2.9 | 46.3±0.3  ${\downarrow}$0.2 |
> | Flickr (ratio = 0.50%)     | 47.1  | 44.1±0.3  ${\downarrow}$3.0 | 46.7±0.2  ${\downarrow}$2.4 |
> | Flickr (ratio = 1.0%)      | 47.1  | 44.7±0.1  ${\downarrow}$2.4 | 46.1±0.1  ${\downarrow}$1.1 |
> | Ogbn-XRT (ratio = 0.05%)   | 55.1  | 52.8±0.4  ${\downarrow}$2.3 | 49.4±0.3  ${\downarrow}$5.8 |
> | Ogbn-XRT (ratio = 0.25%)   | 68.1  | 64.6±0.7  ${\downarrow}$3.5 | 60.7±0.7  ${\downarrow}$7.4 |
> | Ogbn-XRT (ratio = 0.50%)   | 69.3  | 66.1±0.5  ${\downarrow}$3.2 | 63.3±0.4  ${\downarrow}$6.1 |
> | Ogbn-arxiv (ratio = 0.05%) | 60.2  | 56.6±0.7  ${\downarrow}$3.6 | 55.3±0.4  ${\downarrow}$4.9 |
> | Ogbn-arxiv (ratio = 0.25%) | 63.4  | 59.3±0.6  ${\downarrow}$4.1 | 62.7±0.7  ${\downarrow}$0.7 |
> | Ogbn-arxiv (ratio = 0.50%) | 64.8  | 60.9±0.5  ${\downarrow}$3.9 | 63.4±0.5  ${\downarrow}$1.4 |
> | Reddit (ratio = 0.01%)     | 88.0  | 85.9±1.1  ${\downarrow}$2.1 | 80.9±1.0  ${\downarrow}$7.1 |
> | Reddit (ratio = 0.05%)     | 89.6  | 84.8±0.7  ${\downarrow}$5.2 | 83.9±0.9  ${\downarrow}$6.4 |
> | Reddit (ratio = 0.50%)     | 90.1  | 85.3±0.6  ${\downarrow}$4.8 | 86.6±0.9  ${\downarrow}$3.5 |

---

> ### Author Response · Authors · 2023-11-14
> **Response to Reviewer kSU8(2/2)**
>
> **Weaknesses 2: a regularization term based on the Euclidean distance is somewhat inconsistent with their statement and might be misleading, as the Euclidean distance also considers differences beyond just the magnitude of vectors.**
>
> **A2:**
>
> We appreciate your attention to detail on this point.
>
> To sum up, the role of the Euclidean distance is not that of a regularization term. Rather, it serves as **a crucial metric** for assessing gradient differences and is instrumental in fitting frequency domain distributions. And we don't find sufficient literature demonstrating that the Euclidean distance captures information beyond gradient distances.
>
> Firstly, the CTRL method represents an enhancement of existing gradient matching approaches. The role of Euclidean distance is to collaborate with cosine distance for **improved gradient measurement and matching in the frequency domain**. It is not merely a regularization term but rather serves as a means to better quantify gradients and perform matching in the frequency domain. The utilization of Euclidean distance in the improvement of previously researched gradient matching methods based solely on cosine distance is not contradictory to our claims, and it does not lead to any misleading interpretations. Secondly, we acknowledge that Euclidean distance, as a **widely recognized method for measuring gradient magnitude**, raises questions about the additional information it considers. We opted for using Euclidean distance to measure gradient magnitude because our preliminary experiments demonstrated a strong correlation between the gradient magnitude measured by Euclidean distance and the frequency domain distribution of the graph. Many studies involving the measurement of gradient magnitude employed Euclidean distance without explicit evidence of considering additional information. While experimental results do indicate that using Euclidean distance can yield better results than MSE distance, this is largely attributed to the superior ability of Euclidean distance to measure the magnitude of gradients effectively.
>
> **Weaknesses 3: From the presented results, it seems that the proposed method does not offer significant improvements over the compared state-of-the-art.**
>
> **A3:**
>
> In comparison to GCond and DosCond, the primary baselines for gradient matching, CTRL demonstrates average improvements of approximately **2.4% and 1.9%**, with maximum improvements reaching **6% and 6.2%**, respectively.
>
> Notably, even when compared to the latest method SFGC, CTRL still achieves superior results, while SFGC exhibits excellent performance in node classification, it incurs significant computational costs. For instance, the computation time for the buffer may even exceed that used in the distillation process. In contrast, CTRL achieves **better results at a lower cost**. Furthermore, it is crucial to highlight that the SFGC method lacks the capability, as demonstrated by CTRL, to perform dataset distillation in graph classification tasks. These aspects underscore the significance of the CTRL method in the field of graph dataset distillation.
>
> **Q1: instead of using the proposed Euclidean distance between gradients**$||G_i^S - G_i^T||$**, why not directly employ**$||G_i^S|| - ||G_i^T||$
>
> **A4:**
>
> Thanks a lot for your attention!
>
> In brief, $||G_i^S - G_i^T||$can provide **a more nuanced measurement** of the difference between two gradient tensors on each dimension.
>
> On the other hand, the computation method$||G_i^S|| - ||G_i^T||$treats gradients as **scalars** before calculating, representing a relatively **coarser granularity**. We propose utilizing weighted Cosine and Euclidean distance for gradient matching to finely gauge gradient differences while preserving the frequency domain distribution information of the original graph as much as possible. Using coarse-grained computation methods would result in significant information loss in gradients.
>
> ---
>
> [A] Wei Jin, Xianfeng Tang, Haoming Jiang, Zheng Li, Danqing Zhang, Jiliang Tang, Bing Yin: Condensing Graphs via One-Step Gradient Matching. KDD 2022: 720-730
>
> [B] Yanqing Liu, Jianyang Gu, Kai Wang, Zheng Zhu, Wei Jiang, Yang You: DREAM: Efficient Dataset Distillation by Representative Matching. CoRR abs/2302.14416 (2023)

---

> ### Author Response · Authors · 2023-11-17
>
> Thank you for your constructive suggestions on the paper. If you have any further questions, we are more than willing to provide clarifications and conduct additional experiments. We look forward to hearing from you soon.

---

> > ### Comment · Reviewer_kSU8 · 2023-11-21
> >
> > Thank you for your response. Here are some remaining concerns regarding your reply to some of the weaknesses or questions I raised.
> >
> > Weakness 1: I have acknowledged your contribution regarding the correlation of optimizing the Euclidean distance of gradients and frequency distribution alignment, as mentioned in the strength I raised. However, not just me, but other reviewers as well, generally perceive your method as a seemingly simple adaptation of an existing dataset distillation method introduced for CV literature. Additionally, your response still lacks clarity on how your method's initialization using K-means differs from the initialization used in Dream.
> >
> > Weakness 2 & Question 1: In your paper, you stress the correlation between matching gradient magnitude and frequency distribution alignment. However, it seems you've mainly shown a correlation between optimizing the Euclidean distance of gradients and frequency distribution alignment. As I've previously noted, Euclidean distance reflects both magnitude and directional differences between vectors. Given this, why also incorporate cosine similarity instead of exclusively using Euclidean distance? Furthermore, the results in Figure 3(e) are somewhat unexpected. It looks like the accuracy is generally higher when cosine similarity is emphasized (smaller $\beta$) than when Euclidean distance takes precedence (larger $\beta$). This observation raises a question: is optimizing Euclidean distance really as crucial as your paper suggests?

---

> > > ### Author Response · Authors · 2023-11-21
> > > **Response to Reviewer kSU8(2/2)**
> > >
> > > **Weaknesses 2-2: is optimizing Euclidean distance really as crucial as your paper suggests?**
> > >
> > > Thank you very much for your comments.
> > >
> > >
> > > In summary, smaller$\beta$**does not diminish the importance of optimizing Euclidean distance**.
> > >
> > >
> > > Instead, it serves the purpose of preventing the optimization of Euclidean distance from interfering with the optimization of cosine distance in situations where cosine distance is small and Euclidean distance is large. We have empirically demonstrated that neglecting Euclidean distance or considering only Euclidean distance both leads to suboptimal results.
> > >
> > >
> > > To be more specific, relying solely on Euclidean distance as a metric has indeed shown superior performance on some datasets compared to cosine distance methods. However, as illustrated in the table below, our approach consistently outperforms using either cosine or Euclidean distance alone in all experiments.
> > >
> > >
> > > Secondly, the reason why smaller values of $\beta$can lead to better performance is that when optimizing cosine distance, the Euclidean distance between gradients may still be of a considerable magnitude, as depicted in Figure 3: (a) and (b). Setting a larger$\beta$value may impact the optimization of cosine distance.
> > >
> > >
> > > Note that the experiments are conducted under equivalent initialization settings, and all results in the table are averaged over three repetitions for accuracy and consistency.
> > > |  | Ratio | CTRL | GCond(cos) | GCond(euclidean) |
> > > | --- | --- | --- | --- | --- |
> > > | Cora  | 1.30% | 81.9($\beta$ = 0.7) | 79.9${\downarrow}$2.0 | 79.2${\downarrow}$2.7 |
> > > | Cora  | 2.60% | 81.8($\beta$ = 0.05) | 80.0${\downarrow}$1.8 | 80.9${\downarrow}$0.9 |
> > > | Cora  | 5.20% | 81.8($\beta$ = 0.15) | 79.1${\downarrow}$2.7 | 79.3${\downarrow}$2.5 |
> > > | Citeseer | 0.90% | 73.3($\beta$ = 0.9) | 70.7${\downarrow}$2.6 | 71.7${\downarrow}$1.6 |
> > > | Citeseer | 1.80% | 73.5($\beta$ = 0.9) | 70.9${\downarrow}$2.6 | 72.4${\downarrow}$1.1 |
> > > | Citeseer | 3.60% | 73.4($\beta$ = 0.9) | 70.3${\downarrow}$3.1 | 72.7${\downarrow}$0.7 |
> > > | Flickr | 0.10% | 47.1($\beta$ = 0.7) | 46.5${\downarrow}$0.6 | 45.9${\downarrow}$0.2 |
> > > | Flickr | 0.50% | 47.4($\beta$ = 0.05) | 47.1${\downarrow}$0.3 | 46.8${\downarrow}$0.6 |
> > > | Flickr | 1.0% | 47.5($\beta$ = 0.15) | 47.1${\downarrow}$0.4 | 47.0${\downarrow}$0.5 |
> > > | Ogbn-XRT | 0.05% | 61.1($\beta$ = 0.3) | 55.1${\downarrow}$6.0 | 53.2${\downarrow}$7.9 |
> > > | Ogbn-XRT | 0.25% | 69.4($\beta$ = 0.15) | 68.1${\downarrow}$1.3 | 66.5${\downarrow}$2.9 |
> > > | Ogbn-XRT | 0.50% | 70.4($\beta$ = 0.3) | 69.3${\downarrow}$1.1 | 68.9${\downarrow}$1.5 |
> > > | Ogbn-arxiv | 0.05% | 64.9($\beta$ = 0.3 ) | 60.2${\downarrow}$4.7 | 58.7${\downarrow}$6.2 |
> > > | Ogbn-arxiv | 0.25% | 66.7($\beta$ = 0.9 ) | 63.4${\downarrow}$3.3 | 64.4${\downarrow}$2.3 |
> > > | Ogbn-arxiv | 0.50% | 67.3($\beta$ = 0.3 ) | 64.8${\downarrow}$2.5 | 64.9${\downarrow}$2.4 |
> > > | Reddit | 0.01% | 88.9($\beta$ = 0.1 ) | 88.0${\downarrow}$0.9 | 85.6${\downarrow}$3.3 |
> > > | Reddit | 0.05% | 90.1($\beta$ = 0.45 ) | 89.6${\downarrow}$0.5 | 87.1${\downarrow}$3.0 |
> > > | Reddit | 0.50% | 91.3($\beta$ = 0.3 ) | 90.1${\downarrow}$1.2 | 88.3${\downarrow}$3.0 |

---

> ### Author Response · Authors · 2023-11-21
> **Response to Reviewer kSU8(1/2)**
>
> **Weakness 1-1: generally perceive your method as a seemingly simple adaptation of an existing dataset distillation method introduced for CV literature.**
>
> Thank you for the feedback.
>
>
> In summary, the incorporation of Euclidean distance in our methodology is **motivated by entirely distinct rationales, effects, and theoretical underpinnings**. The integration of matching gradient magnitudes in GRAPH DD effectively captures the frequency domain distributions of both the original and synthetic graphs.
>
>
> In contrast, such a phenomenon is not observed in CV DD. Our experimental findings also demonstrate that **ostensibly similar approaches can yield significantly divergent outcomes**. Specifically, the direct use of Euclidean distance for gradient matching in CV DD results in adverse effects, while in GRAPH DD, positive effects may be observed at times. Additionally, the use of weighted distances in CV DD leads to an average improvement of approximately **0.2%**, with a maximum improvement of around **1.1%**. In comparison to GCond and DosCond, CTRL exhibits average improvements of about **2.4% and 1.9%**, reaching up to **6% and 6.2%**, respectively.
>
>
> Further elaborations and detailed experimental results are provided in our response to Reviewer C2Qb(1/4). To avoid excessive length in this response, we refrain from reiterating the details here.
>
> **Weakness 1-2: Additionally, your response still lacks clarity on how your method's initialization using K-means differs from the initialization used in Dream.**
>
> Thanks a lot for your reply!
>
> In short, when it comes to the initialization, Dream chooses to select the **center samples** of each clusters. While CTRL randomly selects **one sample** from every sub-cluster.
>
>
> We believe that Dream's method of capturing feature distribution is simple and efficient, but selecting cluster centers as initial values may cause the results to be overly dependent on the accuracy of the initial clustering. If the initial clustering is not accurate enough, then the cluster centers might not be the best representatives(especially on large ratios). Furthermore, considering the differences in the graph data, such as boundary nodes having edges from multiple classes, correctly classifying these nodes is important for defining decision boundaries. Overemphasis on the choice of centers can lead to a loss of good capture of information from these nodes. Random selection in sub-clusters can mitigate this dependency to some extent, as it is not strictly based on the geometric center of the cluster.
>
> **Weaknesses 2-1: Euclidean distance reflects both magnitude and directional differences between vectors**
>
> Thanks a lot for your significant question!
>
>
> Euclidean distance indeed reflects both magnitude and directional differences between vectors. However, when the angular difference between vectors is small, the Euclidean distance **primarily measures amplitude differences**, as demonstrated in the proof below.
>
>
> Let's assume two vectors,  $\vec{a}$ and $\vec{b}$, with magnitudes$a$and$b$, respectively, and an angle$\theta$between them. The Euclidean distance is given by:
> $$d(\vec{a},\vec{b}) = \sqrt{(\vec{a}-\vec{b})\cdot(\vec{a}-\vec{b})}$$
> Expanding this expression using the properties of vector dot product:
> $$d(\vec{a},\vec{b}) = \sqrt{a^2+b^2-2ab\cos\theta}$$
> During the gradient matching process, we observed that when optimizing using only cosine distance, the cosine distance between gradients can quickly be optimized to a very small value, indicating that the angular difference between gradients is minimal. However, at the same time, the Euclidean distance between these gradients remains large and shows no signs of optimization. This phenomenon is illustrated in Figure 3(a) and (b), where the Euclidean distance difference mainly arises from the amplitude differences between gradients when the angular difference is small.
> In this case, we can use a Taylor series expansion:
> $$\cos\theta \approx 1-\frac{\theta^2}{2}$$
> Substituting this approximation into the expression:
> $$d(\vec{a},\vec{b}) \approx \sqrt{a^2+b^2-2ab(1-\frac{\theta^2}{2})}$$
> Simplifying further:
> $$d(\vec{a},\vec{b}) \approx \sqrt{(a-b)^2+ab\theta^2} \approx \sqrt{(a-b)^2}$$
> As $\theta$ is very small, $(a-b)^2$ dominates the Euclidean distance measurement, indicating that, in this case, the Euclidean distance primarily measures the amplitude differences between the two vectors.

---

> ### Author Response · Authors · 2023-11-22
> **Response to Reviewer kSU8**
>
> We greatly value the constructive suggestions you have offered, and we have summarized the responses and look forward to further discussion with you.
>
> Q1: generally perceive your method as a seemingly simple adaptation of an existing dataset distillation method introduced for CV literature.
>
> A1: In summary, the incorporation of Euclidean distance in our methodology is **motivated by entirely distinct rationales, effects, and theoretical underpinnings**. The integration of matching gradient magnitudes in GRAPH DD effectively captures the frequency domain distributions of both the original and synthetic graphs.
>
>
> Q2: Additionally, your response still lacks clarity on how your method's initialization using K-means differs from the initialization used in Dream.
>
> A2: In short, when it comes to the initialization, Dream chooses to select **the center samples** of each clusters. While CTRL **randomly selects one sample** from every sub-cluster.
>
> Q3: Euclidean distance reflects both magnitude and directional differences between vectors
>
>
> A3: Euclidean distance indeed reflects both magnitude and directional differences between vectors. However, **when the angular difference between vectors is small**, the Euclidean distance **primarily measures amplitude differences**.
>
> Q4: is optimizing Euclidean distance really as crucial as your paper suggests?
>
> A4: In summary, **smaller $\beta$ does not diminish the importance of optimizing Euclidean distance**.

---

### Official Review · Reviewer_ftx3 · 2023-10-31

**Soundness:** 2 fair
**Presentation:** 2 fair
**Contribution:** 2 fair
**Rating:** 5
**Confidence:** 4

**Summary:**

This paper studies the weaknesses of previous graph condensation methods, specifically GCOND, which aims to synthetic a small graph to replace the original graph by matching model gradients. This paper points out two shortcomings: First, GCOND uses cosine distance to measure the similarity of gradients, ignoring the magnitude of gradients. Second, the random initialization of GCOND results in poor diversity. To improve the performance, this paper introduces Euclidean distance and clustering-based sampling, highlighting that they can preserve the frequency distribution of the original graph. Extensive experiments demonstrate the effectiveness of the proposed method.

**Strengths:**

1. This paper presents a new perspective on the relationship between gradient matching methods and the frequency distribution of signals.

2. The extensive experiments, including performance, generalization, visualization, and neural architecture search, validate the effectiveness of the proposed method. Besides, the proposed method achieves state-of-the-art performances in most cases.

**Weaknesses:**

1. The Euclidean distance has been widely used in the trajectory matching [1]. Replacing the cosine distance with Euclidean distance in gradient matching is not very surprising.

2. The empirical results of this paper show that minimizing the Euclidean distance between model gradients can align the frequency distribution between synthetic and real graphs. However, there is no theoretical analysis of the relationship between gradient magnitude and frequency distribution. Additionally, it is unclear to me why matching the frequency distribution of graphs helps the distillation process.

3. There is no ablation study to evaluate the roles of direction-based matching, magnitude-based matching, and their combination.

4. This paper does not report of performance of SFGC [2], which is a strong graph condensation baseline, in Tables 3 and 4.

5. Minor Typo. In several instances, the authors mistakenly write ‘Ogbn-arxiv’ as ‘Ogbn-arvix’.

[1] Dataset Distillation by Matching Training Trajectories. CVPR 2022

[2] Structure-free graph condensation: From large-scale graphs to condensed graph-free data. NeurIPS 2023

**Questions:**

See weaknesses.

---

> ### Author Response · Authors · 2023-11-14
> **Response to Reviewer ftx3(1/3)**
>
> Thanks a lot for your affirmation of our method and performance! We sincerely thank you for the detailed comments and insightful questions. We will respond to your comments as follows.
>
>
>
> **Weaknesses 1: The Euclidean distance has been widely used in the trajectory matching. Replacing the cosine distance with Euclidean distance in gradient matching is not very surprising.**
>
> **A1:**
>
> To conclude, as a parameter-matching method, trajectory matching is completely different from our scheme. The introduction of Euclidean distance in the gradient matching approach **stems from entirely different considerations**. Specifically, it takes into account the **vector nature** of gradients and addresses the necessity of **fitting frequency domain distributions**. First, the trajectory matching method MTT[A] trains many expert models and then matches the training trajectories of the expert models with the trajectories generated by training models with synthetic data, which is done by reducing the Euclidean distance between the **parameters** of the models. This choice of using Euclidean distance is natural as the parameter space of the model does not have directionality like gradients do. It is important to note that our method is based on **gradients** generated by training the model using original and synthetic data.
>
> As vectors, previous research mainly considered matching the direction of gradients while ignoring the factor of gradient magnitude. We start with the natural properties of vectors[B] and the imitation of structural similarity in the frequency domain of graphs, and propose incorporating the match of the Euclidean distance. This is completely different from the method in MTT that uses Euclidean distance to measure parameter similarity. At the same time, we did not propose to replace the cosine distance with the Euclidean distance. As shown in Formula 3 in the paper, we combined the cosine distance and Euclidean distance to perform a more fine-grained gradient difference measurement to achieve a finer gradient matching. The experiment also proved that our method can better preserve the frequency domain distribution information of the graph.
>
>
>
> **Weaknesses 2: it is unclear to me why matching the frequency distribution of graphs helps the distillation process.**
>
> **A2:**
>
> In summary, due to the **strong correlation** between gradient magnitudes and the frequency domain distribution of the graph, it is possible to indirectly fit the frequency domain distribution by optimizing gradient magnitudes. Our experimental results also **validate this assumption**.
>
> Existing research has already demonstrated the significant impact of both frequency domain distribution and graph structure in graph datasets on the performance of Graph Neural Networks (GNNs) [C][D]**.** This motivates us to investigate whether enhancing the distillation of datasets can be achieved by fitting the frequency domain information of synthesized graphs to that of the original graphs.
>
> However, explicitly optimizing graph structural information in the domain of graph data generation is particularly challenging because of the substantial differences in scale between the two and the matrix decomposition involved in many indicators of frequency distribution, making it a computationally intensive process. In our investigation of the correlation between gradient magnitude and the frequency domain distribution of graphs, we found a strong correlation between the two, giving rise to the idea of utilizing gradient magnitude matching to fit frequency distributions.
>
> The core rationale behind matching the frequency distribution lies in the strong correlation between gradient magnitude and the frequency distribution within the graph. By **optimizing one** of these aspects, it becomes possible to **implicitly enhance the other**. In other words, through matching gradient magnitudes, we guide the synthesis of graphs to possess frequency domain distributions that are closer to those of the original graphs. Our experimental results further affirm that this method leads to **better preservation** of the original graph's structural information, ultimately resulting in improved performance.

---

> ### Author Response · Authors · 2023-11-14
> **Response to Reviewer ftx3(2/3)**
>
> **Weaknesses 3: There is no ablation study to evaluate the roles of direction-based matching, magnitude-based matching, and their combination.**
>
> **A3:**
>
> In brief, the ablation experiments demonstrate that the **combined utilization** of cosine distance and Euclidean distance for gradient matching **surpasses** employing **either of them** individually.
>
> We further supplemented the ablation study to assess the performance differences associated with direction-based matching, magnitude-based matching (including mse-based and norm-based), and CTRL, as illustrated in the table below. Note that all results in the table are based on the same initialization method, i.e. the random true initialization employed in GCond for fair comparison, and all CTRL methods are subjected to the same loss function.
>
> Note that all results in the table are averaged over three repetitions for accuracy and consistency.
>
> |                            | GCond(paper) | GCond(reproduction) | GCond(mse) | GCond(norm) |
> | -------------------------- | ------------ | ------------------- | ---------- | ----------- |
> | Cora (ratio = 1.30%)       | 79.8±1.3     | 79.9±1.1            | 78.4±0.8   | 79.2±0.4    |
> | Cora (ratio = 2.60%)       | 80.1±0.6     | 80.0±0.3            | 72.4±0.5   | 80.8±0.4    |
> | Cora (ratio = 5.20%)       | 79.3±0.3     | 79.1±0.2            | 74.2±0.1   | 79.3±0.6    |
> | Citeseer (ratio = 0.90%)   | 70.5±0.5     | 70.7±0.3            | 64.8±0.5   | 71.7±0.3    |
> | Citeseer (ratio = 1.80%)   | 70.6±0.9     | 70.9±1.0            | 62.3±1.0   | 72.5±0.7    |
> | Citeseer (ratio = 3.60%)   | 69.8±1.4     | 70.3±1.1            | 63.9±0.8   | 72.7±0.4    |
> | Flickr (ratio = 0.10%)     | 46.5±0.4     | 46.6±0.3            | 46.3±0.3   | 45.9±0.2    |
> | Flickr (ratio = 0.50%)     | 47.1±0.1     | 47.1±0.3            | 46.7±0.2   | 46.8±0.1    |
> | Flickr (ratio = 1.0%)      | 47.1±0.1     | 47.1±0.1            | 46.1±0.1   | 47.0±0.1    |
> | Ogbn-XRT (ratio = 0.05%)   | -            | 55.3±0.4            | 49.4±0.3   | 53.2±0.3    |
> | Ogbn-XRT (ratio = 0.25%)   | -            | 68.2±0.7            | 60.7±0.7   | 66.5±0.4    |
> | Ogbn-XRT (ratio = 0.50%)   | -            | 69.1±0.6            | 63.3±0.4   | 68.8±0.3    |
> | Ogbn-arxiv (ratio = 0.05%) | 59.2±1.1     | 60.3±0.6            | 55.3±0.4   | 58.7±0.5    |
> | Ogbn-arxiv (ratio = 0.25%) | 63.2±0.3     | 63.2±0.4            | 62.7±0.7   | 64.4±0.8    |
> | Ogbn-arxiv (ratio = 0.50%) | 64.0±1.4     | 64.9±0.3            | 63.4±0.5   | 64.9±0.4    |
> | Reddit (ratio = 0.01%)     | 88.0±1.8     | 88.3±1.3            | 80.9±1.0   | 85.6±0.6    |
> | Reddit (ratio = 0.05%)     | 89.6±0.7     | 89.2±0.8            | 83.9±0.9   | 87.1±0.7    |
> | Reddit (ratio = 0.50%)     | 90.1±0.5     | 90.1±0.7            | 86.6±0.9   | 88.3±0.6    |
>
> **Weaknesses 4: This paper does not report the performance of SFGC in Table 3 and 4.**
>
> **A4:**
>
> Thank you very much for your insightful suggestion regarding the inclusion of SFGC in our work.
>
> SFGC indeed serves as a robust baseline, demonstrating outstanding performance in tasks such as node classification. However, we did not compare our approach with SFGC in Table 3 and Table 4, primarily due to the following considerations:
>
> 1. It follows a fundamentally **distinct technological trajectory** based on parameter matching compared to CTRL, which relies on gradient matching. Much like the disparity between IDC and MTT in CV DD, the commendable cross-architecture performance of SFGC is predicated on extensive time and memory-consuming training of expert models. Given equivalent computational and memory resources, a gradient matching-based approach can feasibly distill corresponding synthetic datasets for each model architecture.
> 2. SFGC didn't present experimental results for the Versatility experiment in Table 4 and the Neural Architecture Search experiment in Table 5. Owing to constraints in time and computational resources, we are unable to undertake a corresponding reproduction for comparative purposes.
>
>
>
> **Weaknesses 5: Minor Typo**
>
> **A5:**
>
> We have carefully reviewed the entire manuscript and corrected all typos. Thank you again for your meticulous review and comments!
>
> ---
>
> [A] George Cazenavette et.al. Dataset Distillation by Matching Training Trajectories. CVPR 2022.
>
> [B] Ellen H Fukuda et.al. On the convergence of the projected gradient method for vector optimization. Optimization (page1009-1021).
>
> [C] Deyu Bo et.al. Beyond Low-frequency Information in Graph Convolutional Networks. AAAI 2021.
>
> [D] Ruiyi Fang et al. Structure-Preserving Graph Representation Learning. ICDM2022.

---

> ### Author Response · Authors · 2023-11-17
>
> We highly appreciate the insightful suggestions you provided. We have taken them into account and conducted experiments accordingly. Should there be any remaining queries or if you wish for more experiments, we are eager to proceed. We look forward to your response.

---

> ### Author Response · Authors · 2023-11-19
> **Response to Reviewer ftx3(3/3)**
>
> **Weaknesses 2: there is no theoretical analysis of the relationship between gradient magnitude and frequency distribution.**
>
> **A5:**
>
> In summary, the sensitivity of graph neural networks (GNNs) to the frequency-domain structure of input graphs during training arises from **the impact of different distributions of high and low-frequency signals on convolution and deconvolution operations**.
>
> The training process of graph neural networks involves frequency-domain convolution and deconvolution, where graph signals and error signals are decomposed into linear combinations of different frequency components using the eigenvectors of the graph Laplacian matrix as Fourier bases. The correlation between gradient magnitude and the distribution of high and low-frequency signals in the input graph can be attributed to two main factors. Firstly, convolution and deconvolution operations lead to spectral bias, wherein the network tends to preserve low-frequency signals, affecting gradient magnitude due to spectral shift. Secondly, spectral filtering induced by convolution and deconvolution further influences gradient magnitude by weighting different frequency components of signals using the Laplacian matrix's eigenvectors.
>
> In specific terms, the training process of graph neural networks can be viewed as performing convolution and deconvolution in the frequency domain. The forward pass of GNNs transforms graph signals from spatial to frequency domain, performs convolution in the frequency domain, and then returns to the spatial domain. The backward pass transforms error signals from spatial to frequency domain, performs deconvolution in the frequency domain, and then returns to the spatial domain. The core of convolution and deconvolution lies in utilizing the eigenvectors of the graph Laplacian matrix as Fourier bases to decompose graph signals and error signals into different frequency components.
>
> During the training of graph neural networks, the strong correlation between gradient magnitude and the distribution of high and low-frequency signals in the input graph can be explained by two main reasons. Firstly, **convolution and deconvolution operations in GNNs cause spectral bias**, where the network tends to preserve low-frequency signal components and suppress high-frequency ones. Therefore, when the distribution of high and low-frequency signals in the input graph is imbalanced, these operations alter the spectra of graph signals and error signals, impacting gradient magnitude. For instance, when the input graph has more high-frequency signal components, convolution weakens the high-frequency components of graph signals, while deconvolution strengthens the high-frequency components of error signals, leading to an increase in gradient magnitude. Conversely, when the input graph has more low-frequency signal components, convolution preserves the low-frequency components of graph signals, and deconvolution weakens the low-frequency components of error signals, resulting in a decrease in gradient magnitude.
>
> Secondly, **convolution and deconvolution operations in GNNs cause spectral filtering**, where the convolution kernel weights different frequency components of graph signals and error signals. This is because the convolution kernel in GNNs is a function of the eigenvalues of the graph Laplacian matrix, acting as a frequency-domain filter that amplifies or attenuates different frequency components of graph signals and error signals. Therefore, when the distribution of high and low-frequency signals in the input graph is asynchronous, the convolution kernel has varying effects on the spectra of graph signals and error signals, influencing gradient magnitude. For example, when the input graph has more high-frequency signal components and the convolution kernel attenuates high-frequency components, convolution weakens the high-frequency components of graph signals, and deconvolution strengthens the high-frequency components of error signals, resulting in an increase in gradient magnitude. Conversely, when the input graph has more low-frequency signal components and the convolution kernel amplifies low-frequency components, convolution strengthens the low-frequency components of graph signals, and deconvolution weakens the low-frequency components of error signals, leading to a decrease in gradient magnitude.

---

> ### Author Response · Authors · 2023-11-21
> **Official Comment by Authors**
>
> We greatly value the constructive suggestions you have offered, and we have diligently incorporated them into our work, conducting experiments as per your recommendations. If there are any lingering queries or if you desire further experiments to address specific aspects, we are more than willing to undertake them. Your feedback is instrumental in enhancing the quality of our research, and we eagerly anticipate your continued guidance.

---

> > ### Author Response · Authors · 2023-11-22
> > **Response to Reviewer ftx3**
> >
> > We greatly value the constructive suggestions you have offered, and we have summarized the responses and look forward to further discussion with you.
> >
> >
> > Q1: The Euclidean distance has been widely used in the trajectory matching.
> >
> >
> > A1:
> > To conclude, as a parameter-matching method, trajectory matching is completely different from our scheme. The introduction of Euclidean distance in the gradient matching approach **stems from entirely different considerations**. Specifically, it takes into account the vector nature of gradients and addresses the necessity of fitting frequency domain distributions.
> >
> >
> > Q2: It is unclear why matching the frequency distribution of graphs helps the distillation process.
> >
> >
> > A2:
> > In summary, due to the **strong correlation** between gradient magnitudes and the frequency domain distribution of the graph, it is possible to indirectly fit the frequency domain distribution by optimizing gradient magnitudes. Our experimental results also validate this assumption.
> >
> > Q3: There is no ablation study to evaluate the roles of direction-based matching, magnitude-based matching, and their combination
> >
> >
> > A3:
> > In brief, the ablation experiments demonstrate that the **combined utilization** of cosine distance and Euclidean distance for gradient matching **surpasses employing either of them individually**.
> >
> > Q4:  there is no theoretical analysis of the relationship between gradient magnitude and frequency distribution.
> >
> >
> > A4:
> > In summary, the sensitivity of graph neural networks (GNNs) to the frequency-domain structure of input graphs during training arises from **the impact of different distributions of high and low-frequency signals on convolution and deconvolution operations**.
> >
> > Q5: This paper does not report the performance of SFGC in Table 3 and 4.
> >
> >
> > A5:
> > SFGC follows a fundamentally **distinct technological trajectory** based on parameter matching compared to CTRL, which relies on gradient matching. Much like the disparity between IDC and MTT in CV DD, the commendable cross-architecture performance of SFGC is predicated on extensive time and memory-consuming training of expert models. Given equivalent computational and memory resources, a gradient matching-based approach can feasibly distill corresponding synthetic datasets for each model architecture.

---

### Official Review · Reviewer_x24f · 2023-11-01

**Soundness:** 3 good
**Presentation:** 3 good
**Contribution:** 3 good
**Rating:** 6
**Confidence:** 3

**Summary:**

This paper studies a newly proposed graph condensation method. The authors argue that existing methods focus solely on matching gradient directions, which can lead to issues. To address this, the authors propose CTRL, a novel method that incorporates gradient magnitude matching using Euclidean distance and ensures an even feature distribution. CTRL outperforms existing methods in experiments on multiple datasets, making it a promising addition to graph distillation techniques.

**Strengths:**

1. It is crucial to address the issue of existing costs in training large-scale graphs through graph condensation and summarization, as it remains an under-explored problem.
2. It is interesting to observe how the authors have justified a strong correlation between the frequency distribution and the gradient magnitude during training GNN models.
3. Various experiments are conducted, including the generalization experiments

**Weaknesses:**

1. In section A.2, it is stated, 'We match gradients smaller than this threshold on both direction and magnitude, while gradients exceeding the threshold are matched solely based on their directions.' Can the authors provide a detailed demonstration of why considering both gradient matching based on direction and magnitude may not be helpful? This is essential for assessing the effectiveness of CTRL for large datasets, and it appears to be missing from the conducted experiments.
2. The overall method is similar to the recently proposed graph condensation methods (especially GCond as it also uses a gradient matching scheme). The novelty is somewhat limited, although the struggle to capture the feature distribution in this problem is interesting.
3. Can the authors explain in detail why leveraging gradient magnitude matching is correlated with frequency distribution? Did the authors also measure the correlation between gradient direction matching (cosine similarity between gradients) and frequency distribution? These correlations can be investigated through running some experiments.

**Questions:**

My major questions on the experimental evaluation. It would tremendously strengthen this work by addressing the concerns listed in the Weakness section.

---

> ### Author Response · Authors · 2023-11-14
> **Response to Reviewer x24f(1/3)**
>
> Thank you very much for your recognition and affirmation of our method and experiments!
>
> We sincerely thank you for the detailed comments and insightful questions. We will respond to your comments as follows.
>
>
>
> **Weaknesses1: Can the authors provide a detailed demonstration of why considering both gradient matching based on direction and magnitude may not be helpful?**
>
> **A1:**
>
> Thanks for the comment.
>
> In conclusion, utilizing a threshold is an empirical approach. Its adoption **does not significantly impact performance** but rather affects runtime and the eventual choice of the $\beta$ value.
>
> In [A], it is mentioned that the cosine-based matching method in Data Distillation of Computer Vision(CV DD) may fail when the gradient values are close to zero. Although this situation did not occur in our actual training scenarios (the gradient value of the training GNN on the synthetic data set did not decrease close to 0), we conducted corresponding experiments. Specifically, we straightforwardly calculated the gradient magnitudes generated during each gradient matching process from the synthetic dataset training. We then selected approximately 20% of the boundary points as the threshold (Ogbn-arxiv: 50, Reddit: 10).
>
> It is important to note that we did not meticulously tune this threshold as a hyperparameter, rather, it was an attempt to expedite training and enhance effectiveness. Moreover, its presence or absence has a minimal impact on performance. The results below demonstrate the corresponding outcomes with and without the threshold (we only applied the threshold on Ogbn-arxiv and Reddit datasets).
>
> An intriguing observation lies in the introduction of a threshold, where the optima l$\beta$ value tends to be relatively small. However, **setting it to zero also results in suboptimal outcomes**. This phenomenon may be attributed to the fact that when gradients generated from synthetic data are small, the measure of direction difference becomes more crucial than the magnitude difference. Therefore, assigning a greater weight to direction difference is necessary. Additionally, the prohibition of​ $\beta$ being set to zero further underscores the significance of introducing Euclidean distance in gradient matching.
>
> Note that all results in the table are averaged over three repetitions for accuracy and consistency.
>
> |                            | CTRL(w/o threshold)                        | CTRL(w/ threshold)   |
> | - | - | - |
> | Ogbn-arxiv (ratio = 0.05%) | 65.2($\beta$ = 0.3 )     $\downarrow$0.4 | 65.6($\beta$ = 0.05) |
> | Ogbn-arxiv (ratio = 0.25%) | 66.7($\beta$ = 0.9 )  $\uparrow$0.2        | 66.5($\beta$ = 0.05) |
> | Ogbn-arxiv (ratio = 0.50%) | 67.2($\beta$ = 0.3 )     $\downarrow$0.4 | 67.6($\beta$ = 0.15) |
> | Reddit (ratio = 0.01%)     | 89.3($\beta$ = 0.1 )  $\uparrow$0.1      | 89.2($\beta$ = 0.1)  |
> | Reddit (ratio = 0.05%)     | 90.2($\beta$ = 0.45 )$\downarrow$0.4      | 90.6($\beta$ = 0.2)  |
> | Reddit (ratio = 0.50%)     | 91.5($\beta$ = 0.3 )$\downarrow$0.5       | 91.9($\beta$ = 0.1)  |

---

> ### Author Response · Authors · 2023-11-14
> **Response to Reviewer x24f(2/3)**
>
> **Weaknesses2：The overall method is similar to the recently proposed graph condensation methods (especially GCond as it also uses a gradient matching scheme)**
>
> **A2:**
>
> To sum up, gradient matching is a commonly employed technique, and it has gradually become outdated in the field of CV DD. Our approach not only **reveals the shortcomings** of the current criterion of gradient matching in the realm of GRAPH DD from a frequency domain perspective, but also demonstrates **the further research value** of gradient matching in GRAPH DD.
>
> In Data Distillation, gradient matching is a very general scheme. In CV DD, DC[B] is one of the earliest methods based on gradient matching. Many other methods, such as DSA[C] and IDC[D], are improved methods based on DC. Similarly, we explore further to improve the gradient matching approach in graph condensation.
>
> To delineate our contributions more precisely, we can categorize them into three main facets:
>
> 1. Unlike previous studies, we **depart from** **frequency domain distribution** to investigate the relationship between gradient magnitude and graph structure. Furthermore, we elaborate on the **rationale** behind introducing Euclidean distance in gradient matching. It is noteworthy that we implemented similar enhancements on the IDC, but the improvement was not nearly as significant as observed in the graph dataset distillation. For instance, the average improvement is about 0.2% and the maximum improvement is about 1.1%, while compared to GCond and DosCond, the average improvements of CTRL are about 2.4% and 1.9%, up to 6% and 6.2%, respectively.
>
> 2. In the realm of CV DD, the advent of the parameter-matching method MTT[E] has seemingly rendered gradient matching outdated. However, our research effectively demonstrates that in the GRAPH DD domain, even in the absence of the time and space-consuming buffer process, a gradient-matching approach can outperform parameter-matching methods by incorporating frequency domain distribution matching. Our research demonstrates **the potential of gradient-based methods in the GRAPH DD domain**.
>
> 3. Additionally, many initialization methods commonly employed in CV DD cannot be directly applied in GRAPH DD. For example, the selection of images that can be correctly classified may not be directly transferable to the context of graph datasets, while the current initialization methods within the realm of graph compression are overly simplistic. Therefore, we propose an initialization method that can capture the feature distribution of original data, and provides a better optimization starting point for synthetic graph while introducing almost no additional computational cost. This is an area that **has not been explored in previous research**.
>
>
> We demonstrate the effectiveness of CTRL's two components through extensive experimentation. Additionally, the two approaches are easy to follow and can be effortlessly integrated into other gradient-matching-based graph condensation methods.

---

> ### Author Response · Authors · 2023-11-14
> **Response to Reviewer x24f(3/3)**
>
> **Weaknesses 3-1: Can the authors explain in detail why leveraging gradient magnitude matching is correlated with frequency distribution?**
>
> **A3:**
>
> In brief, due to the **strong correlation** between gradient magnitudes and the frequency domain distribution of the graph, it is possible to indirectly fit the frequency domain distribution by optimizing gradient magnitudes. Our experimental results also **validate this assumption**.
>
> Existing research has already demonstrated the significant impact of both frequency domain distribution and graph structure in graph datasets on the performance of Graph Neural Networks (GNNs) [F][G]. This motivates us to investigate whether enhancing the distillation of datasets can be achieved by fitting the frequency domain information of synthesized graphs to that of the original graphs.
>
> However, explicitly optimizing graph structural information in the domain of graph data generation is particularly challenging because of the substantial differences in scale between the two and the matrix decomposition involved in many indicators of frequency distribution, making it a computationally intensive process. In our investigation of the correlation between gradient magnitude and the frequency domain distribution of graphs, we found a strong correlation between the two, giving rise to the idea of utilizing gradient magnitude matching to fit frequency distributions.
>
> The core rationale behind leveraging gradient magnitude matching lies in the strong correlation between gradient magnitude and the frequency distribution within the graph. By **optimizing one** of these aspects, it becomes possible to **implicitly enhance the other**. In other words, through matching gradient magnitudes, we guide the synthesis of graphs to possess frequency domain distributions that are closer to those of the original graphs. Our experimental results further affirm that this method leads to **better preservation** of the original graph's structural information, ultimately resulting in improved performance.
>
>
>
> **Weaknesses 3-2: Did the authors also measure the correlation between gradient direction matching (cosine similarity between gradients) and frequency distribution?**
>
> **A4:**
>
> To sum up, experimental results demonstrate that the variation in gradient direction exhibits **a much lower correlation**  (average of 0.569 compared to an average of 0.905)  with the frequency domain distribution of the graph compared to the gradient magnitudes.
>
> If the question referring to whether matching gradients can preserve certain frequency domain information, i.e., whether the synthetic images generated by the GCond method can exhibit frequency distribution similar to that of the original images, as shown in Table 6, it may achieve this to some extent, but not as effective as our proposed method.  We conducted additional experiments and found **a relatively weak correlation between gradient direction and frequency domain distribution**, compared to the result in Table 9. This also explains why the gradient matching method based on cosine distance can also generate a composite graph with some information in the frequency domain of the original graph.
>
> The table below shows the Spearman correlation coefficients of the high-frequency area and gradient direction under different graph models and GNN architectures, where the gradient direction is measured by the variance of an array of cosine distance between gradients of the previous epoch and the next epoch.
>
> |                 | SGC   | Cheby | APPNP | GCN   |
> | - | - | - | - | - |
> | Erdos-Renyi     | 0.675 | 0.755 | 0.513 | 0.614 |
> | Barabasi-Albert | 0.667 | 0.577 | 0.758 | 0.313 |
> | Watts-Strogatz  | 0.386 | 0.422 | 0.489 | 0.661 |
>
> ---
>
> [A] Zixuan Jiang et al. "Delving into effective gradient matching for dataset condensation." 2023 IEEE International Conference on Omni-layer Intelligent Systems (COINS). IEEE, 2023.
>
> [B] Bo Zhao et.al. Dataset Condensation with Gradient Matching. ICLR2021.
>
> [C] Bo Zhao et.al. Dataset Condensation with Differentiable Siamese Augmentation. ICML2021.
>
> [D] Jang-Hyun Kim et.al. Dataset condensation via efficient synthetic data parameterization. ICML2022
>
> [E] George Cazenavette et.al. Dataset distillation by matching training trajectories. CVPR2022.
>
> [F] Deyu Bo et.al. Beyond Low-frequency Information in Graph Convolutional Networks. AAAI 2021.
>
> [G] Ruiyi Fang et al. Structure-Preserving Graph Representation Learning. ICDM 2022.

---

> ### Author Response · Authors · 2023-11-17
>
> We express our gratitude for the reviewer's positive feedback on our work. To further enhance our research, we are more than willing to undertake additional experiments as needed. We are eagerly awaiting your input and insights.

---

> ### Author Response · Authors · 2023-11-21
> **Official Comment by Authors**
>
> We would like to express our sincere appreciation for the valuable insights you have provided. Should there be any remaining questions or if you have specific areas where you believe additional experiments could shed further light, we are enthusiastic about conducting them. Your continued engagement is crucial to refining our contributions, and we eagerly await your feedback.

---

> > ### Comment · Reviewer_x24f · 2023-11-21
> > **Thanks for the rebuttal**
> >
> > Thanks for the detailed rebuttal, and I would like to maintain my score.

---

### Official Review · Reviewer_C2Qb · 2023-11-01

**Soundness:** 3 good
**Presentation:** 3 good
**Contribution:** 2 fair
**Rating:** 5
**Confidence:** 3

**Summary:**

This paper investigates key modules within graph condensation. Firstly, it points out that using cosine similarity as the loss in graph condensation fails to account for gradient magnitude, prompting the adoption of a loss incorporating Euclidean distance. Secondly, the paper highlights that the completely random initialization in graph condensation is suboptimal, leading to the utilization of a clustering algorithm on node features to enhance initialization quality. Extensive experiments demonstrate that the proposed approach improves the quality of the condensed graph and reveals the relationship between graph frequency distribution and gradient magnitude.

**Strengths:**

1. The paper is easy to follow and generally well-written.
2. The motivation appears to be both reasonable and clear.
3. The experiments are thorough and comprehensive.

**Weaknesses:**

1. Novelty of the proposed methods is limited, which seems a simple adaptation of an existing method from CV literature, and the proposed methods do not appear to be specifically designed for graphs. Firstly, the loss formula (Equation 3) that considers gradient magnitude is identical to Equation 9 in [1]. Secondly, the initialization step using k-means is performed on node features, which overlooks the influence of graph structure. For instance, if two nodes have the same features but significantly differ in their neighbors, the approach described in the paper would not select both nodes simultaneously, even though they would have completely different embeddings after message passing.

2. The setup of the ablation experiments is not clearly defined. It is not specified which loss was used to train the models in Figure 3 (c) and (d). It is not specified what initialization was used for the model in Figure 3 (e).

Minor comments:
There are quite a few typos in the paper where "ogbn-arxiv" is misspelled.

[1]Jiang, Zixuan, et al. "Delving into effective gradient matching for dataset condensation." 2023 IEEE International Conference on Omni-layer Intelligent Systems (COINS). IEEE, 2023.

**Questions:**

1. Are there any differences in applying CTRL to graphs compared to its application in computer vision?
2. What is the value of beta in the training of models in Figure 3 (c) and (d)? How is the models in Figure 3 (e) initialized?
3. Some of the compared methods, such as GCond, had their code updated recently. Has the performance of these methods been re-evaluated, specifically regarding the results presented in Table 1?

---

> ### Author Response · Authors · 2023-11-14
> **Response to Reviewer C2Qb(1/4)**
>
> Thanks a lot for the affirmation of our article writing, experiments, and motivation! We sincerely thank you for the detailed comments and insightful questions. We make responses as follows.
>
> **Weaknesses1-1: Novelty of the proposed methods is limited, which seems a simple adaptation of an existing method from CV literature, and the proposed methods do not appear to be specifically designed for graphs**.
>
> **A1：**
>
> We thank the reviewer, and will clarify our technical novelty and significance during revision.
>
> To sum up, the utilization of Euclidean distance in [A] and our approach **stems from entirely distinct motivations, effects, and theoretical foundations.** The introduction of matching gradient magnitudes in GRAPH DD effectively captures the frequency domain distributions of both the original and synthetic graphs. Conversely, such a phenomenon is not observed in [A]. Our experimental results also demonstrate that **seemingly similar approaches can lead to markedly different outcomes.**
>
> First and foremost, a notable distinction from [A] lies in our extended **analysis of the rationale** behind incorporating Euclidean distance in GRAPH DD. Specifically, the frequency domain distribution of the graph encompasses structural information, our experiments show that matching Euclidean distance effectively preserves the high and low-frequency information of the original graph(which is important for GNN training), as shown in Table 6, thereby enhancing the performance of the synthesized dataset.
>
> While in Data Distillation of Computer Vision (CV DD), there is a lack of similar research conclusions. Specifically, [A] proposed a method that introduces Euclidean distance into gradient matching, yielding only a marginal improvement of around 0.5%. In comparison to the improvements introduced in GRAPH DD, the performance is significantly lower. Additionally, we conducted experiments on the state-of-the-art (among those based on gradient matching)  method IDC in CV DD, as shown in the table below, IDC(cos) indicates the original IDC method and IDC(norm) means the IDC method based on Euclidean distance matching.
>
> As shown in the table below, IDC(norm) demonstrated a **significantly lower performance** compared to IDC(cos). Conversely, in GRAPH DD, the Euclidean distance-based matching method, GCond(norm) even exhibited **slightly higher performance** than the cosine distance-based method GCond(cos) in some experiments. This observation further substantiates our claim that by matching gradient magnitudes, we can better approximate the frequency domain distribution of the original graph with synthetic graphs, thereby achieving superior performance.
>
> Note: we use ${\downarrow}$ and ${\uparrow}$ to denote the reduction and increase of Acc, all results in the table are averaged over three repetitions for accuracy and consistency.
>
> |              | IPC  | IDC (cos) | IDC (norm)              |
> | - | - | - | - |
> | MNIST        | 1    | 94.2      | 90.2  ${\downarrow}$4.0 |
> | MNIST        | 10   | 98.4      | 93.2  ${\downarrow}$5.2 |
> | MNIST        | 50   | 99.1      | 94.3  ${\downarrow}$4.9 |
> | FashionMNIST | 1    | 81.0      | 79.6  ${\downarrow}$1.4 |
> | FashionMNIST | 10   | 86.0      | 83.1  ${\downarrow}$2.9 |
> | FashionMNIST | 50   | 86.2      | 84.3  ${\downarrow}$1.9 |
> | SVHN         | 1    | 68.5      | 61.5  ${\downarrow}$7.0 |
> | SVHN         | 10   | 87.5      | 81.4  ${\downarrow}$6.1 |
> | SVHN         | 50   | 90.1      | 87.1  ${\downarrow}$3.0 |
> | CIFAR-10     | 1    | 50.6      | 45.5  ${\downarrow}$4.9 |
> | CIFAR-10     | 10   | 67.5      | 63.5  ${\downarrow}$4.0 |
> | CIFAR-10     | 50   | 74.5      | 72.3  ${\downarrow}$2.2 |
>
> |  | Ratio | GCond(cos) | GCond(norm) |
> | - | - | - | - |
> | Cora  | 1.30% | 79.9 | 79.2${\downarrow}$0.7 |
> | Cora  | 2.60% | 80.0 | 80.9$\uparrow$0.9 |
> | Cora  | 5.20% | 79.1 | 79.3$\uparrow$0.2 |
> | Citeseer | 0.90% | 70.7 | 71.7$\uparrow$1.0 |
> | Citeseer | 1.80% | 70.9 | 72.4$\uparrow$1.5 |
> | Citeseer | 3.60% | 70.3 | 72.7$\uparrow$2.4 |
> | Flickr | 0.10% | 46.5 | 45.9${\downarrow}$0.6 |
> | Flickr | 0.50% | 47.1 | 46.8${\downarrow}$0.3 |
> | Flickr | 1.0% | 47.1 | 47.0${\downarrow}$0.1 |
> | Ogbn-XRT | 0.05% | 55.1 | 53.2${\downarrow}$1.9 |
> | Ogbn-XRT | 0.25% | 68.1 | 66.5${\downarrow}$1.6 |
> | Ogbn-XRT | 0.50% | 69.3 | 68.9${\downarrow}$0.4 |
> | Ogbn-arxiv | 0.05% | 60.2 | 58.7${\downarrow}$1.5 |
> | Ogbn-arxiv | 0.25% | 63.4 | 64.4$\uparrow$1.0 |
> | Ogbn-arxiv | 0.50% | 64.8 | 64.9$\uparrow$0.1 |
> | Reddit | 0.01% | 88.0 | 85.6${\downarrow}$2.4 |
> | Reddit | 0.05% | 89.6 | 87.1${\downarrow}$2.5 |
> | Reddit | 0.50% | 90.1 | 88.3${\downarrow}$1.8 |

---

> ### Author Response · Authors · 2023-11-14
> **Response to Reviewer C2Qb(2/4)**
>
> **Weaknesses1-2: the initialization step overlooks the influence of graph structure.**
>
> **A2:**
>
> Thank you for considering potential issues of the graph structure in our initialization.
>
> In summary, we try to **avoid the inclusion of noisy connections** in synthesized graphs [C], as this is unacceptable for condensed graphs with very small data volumes. Experiments show that considering the graph structure during initialization will **lead to negative gains**.
>
> We do not incorporate the graph structure in initialization but directly capture the feature distribution of the original data for the following reasons :
>
> 1. Firstly, the original graph may contain noise connections, which can greatly deteriorate the performance of GNNs[C]. This is particularly unacceptable for synthetic data, which has only a very small amount of data. To avoid the presence of noise connections, we only focus on the feature distribution during initialization, capturing the feature distribution of the original data. At the same time, we adopted another more reasonable approach to obtain information on graph structure. During the distillation process, we used a structure learner, employing a graph structure learning method based on node features to efficiently obtain graph structure of higher quality.
> 2. Our initialization method aims to cluster the original data using k-means, allowing synthetic data to simulate the real data's feature distribution, rather than aligning with the feature distribution after message passing.
> 3. If we retrain a GNN and use the features processed by GNN message passing as the data for k-means clustering to get the value of initialization, it would introduce the additional cost of training another GNN.
>
> The experimental results are as follows:
>
> Note: we use${\downarrow}$to denote the reduction of Acc of using node features after message passing to the approach adopted by CTRL. We selected the best-performing $\beta$ in the CTRL method for our experiments. All results in the table are averaged over three repetitions for accuracy and consistency.
>
> |                 | Citeseer, ratio=1.80%,$\beta$=0.7 | Cora, ratio=2.60%,$\beta$=0.7 | Ogbn-arxiv, ratio=0.25%,          $\beta$=0.15 | Ogbn-XRT, ratio=0.25%,    $\beta$=0.7 | Flickr, ratio=0.50%,$\beta$=0.1 | Reddit, ratio=0.05%,$\beta$=0.3 |
> | - | - | - | - | - | - | - |
> | Extra cost (s.) | 20±0.5                            | 29±0.6                         | 540±5.3                                        | 560±6.7                               | 45±2.0                          | 2410±10                         |
> | Performance     | 72.1            | 81.1          | 65.9                        | 67.8                | 47.1        | 90.0         |
> | Reduction | ${\downarrow}$1.4 | ${\downarrow}$0.7|${\downarrow}$0.6  |${\downarrow}$1.6 |  ${\downarrow}$0.3 | ${\downarrow}$0.6 |
>
>
> **Q1: Are there any differences in applying CTRL to graphs compared to its application in Computer Vision?**
>
> **A3：**
>
> Thanks for the comment.
>
> In summary, directly using our method in CV DD **does not yield as favorable results** as seen in GRAPH DD. As previously mentioned, the effective rationale behind introducing Euclidean distance in graph DD does not apply in this context.
>
> To further demonstrate the difference between DD in GRAPH and CV yields,  we tried to add CTRL to CV DD, which didn't achieve such good improvement. For instance, we conduct a series of experiments based on two kinds of gradient matching schemes, [A] (We introduce $\beta$) and IDC[B] (We introducing Euclidean distance), the average improvement is about **0.2%** and the maximum improvement is about **1.1%**, while compared to GCond and DosCond, the average improvements of CTRL are about **2.4% and 1.9%**, up to **6% and 6.2%**, respectively.
>
> Note that introducing $\beta$ can slightly improve the method in [A], as shown in the table below, we tried three values of​ $\beta$, 0.1, 0.5, and 0.9, and show the results that have improvement higher than 0.1%. Where MD denotes the method proposed in [A] using both cosine distance and Euclidean distance(equal weight), while MDA is the improved version of MD in [A], all results in the table are averaged over three repetitions for accuracy and consistency.
>
> |              | IPC  | MD       | MD+CTRL                 | MDA      | MDA+CTRL                |
> | - | - | - | - | - | - |
> | FashionMNIST | 10   | 85.4±0.3 | 85.4±0.3                | 84.2±0.3 | 84.7±0.1  $\uparrow$0.5 |
> | FashionMNIST | 50   | 87.4±0.2 | 87.6±0.2  $\uparrow$0.2 | 87.9±0.2 | 87.9±0.2                |
> | SVHN         | 10   | 75.9±0.7 | 75.9±0.4                | 75.9±0.7 | 76.3±0.2  $\uparrow$0.4 |
> | SVHN         | 50   | 82.9±0.2 | 83.0±0.1  $\uparrow$0.1 | 83.2±0.3 | 83.4±0.3  $\uparrow$0.2 |
> | CIFAR-10     | 1    | 30.0±0.6 | 31.0±0.4  $\uparrow$1.0 | -        | -                       |
> | CIFAR-10     | 10   | 49.5±0.5 | 49.5±0.3                | 50.2±0.6 | 51.3±0.6  $\uparrow$1.1 |

---

> ### Author Response · Authors · 2023-11-14
> **Response to Reviewer C2Qb(3/4)**
>
> **Q2：It is not specified which loss was used to train the models in Figure 3 (c) and (d). What is the value of** **$\beta$** **in the training of models in Figure 3 (c) and (d)? How is the models in Figure 3 (e) initialized?**
>
> **A4:**
>
> Thanks for pointing out this specific matter within the paper.
>
> The values of $\beta$ in the training of models in Figure 3 (c) and (d) are 0.1 and 0.15, respectively. We incorporate this information into the article.
>
> In Figure 3 (e), we employ the specific initialization method(K-MEANS-based) in CTRL for Ogbn-arxiv, and random sampling for cora and citeseer, as shown in A.2.
>
> We supplement a complete ablation experiment here, as follows, all the results are obtained by repeating experiments for 3 times.
>
> |                            | GCond(paper) | GCond(reproduction) | GCond+initlization | CTRL w/o initlization    |
> | - | - | - | - | - |
> | Cora (ratio = 1.30%)       | 79.8±1.3     | 79.9±1.1            | 79.4±0.8           | 81.9±0.4($\beta$ = 0.7)  |
> | Cora (ratio = 2.60%)       | 80.1±0.6     | 80.0±0.3            | 80.4±0.6           | 81.8±0.5($\beta$ = 0.05) |
> | Cora (ratio = 5.20%)       | 79.3±0.3     | 79.1±0.2            | 79.2±0.3           | 81.8±0.6($\beta$ = 0.15) |
> | Citeseer (ratio = 0.90%)   | 70.5±0.5     | 70.7±0.3            | 70.8±0.3           | 73.3±0.3($\beta$ = 0.9)  |
> | Citeseer (ratio = 1.80%)   | 70.6±0.9     | 70.9±1.0            | 71.1±1.1           | 73.5±0.8($\beta$ = 0.9)  |
> | Citeseer (ratio = 3.60%)   | 69.8±1.4     | 70.3±1.1            | 70.9±0.9           | 73.4±0.6($\beta$ = 0.9)  |
> | Flickr (ratio = 0.10%)     | 46.5±0.4     | 46.6±0.3            | 46.5±0.3           | 47.1±0.2($\beta$ = 0.7)  |
> | Flickr (ratio = 0.50%)     | 47.1±0.1     | 47.1±0.3            | 46.8±0.3           | 47.4±0.1($\beta$ = 0.05) |
> | Flickr (ratio = 1.0%)      | 47.1±0.1     | 47.1±0.1            | 47.1±0.2           | 47.5±0.1($\beta$ = 0.15) |
> | Ogbn-XRT (ratio = 0.05%)   | -            | 55.3±0.4            | 57.2±0.3           | 61.1±0.3($\beta$ = 0.3)  |
> | Ogbn-XRT (ratio = 0.25%)   | -            | 68.2±0.7            | 69.7±0.5           | 69.4±0.4($\beta$ = 0.15) |
> | Ogbn-XRT (ratio = 0.50%)   | -            | 69.1±0.6            | 69.3±0.5           | 70.4±0.4($\beta$ = 0.3)  |
> | Ogbn-arxiv (ratio = 0.05%) | 59.2±1.1     | 60.3±0.6            | 62.3±0.7           | 64.9±0.7($\beta$ = 0.3)  |
> | Ogbn-arxiv (ratio = 0.25%) | 63.2±0.3     | 63.2±0.4            | 64.7±0.6           | 66.7±0.6($\beta$ = 0.9)  |
> | Ogbn-arxiv (ratio = 0.50%) | 64.0±1.4     | 64.9±0.3            | 65.4±0.7           | 67.3±0.3($\beta$ = 0.3)  |
> | Reddit (ratio = 0.01%)     | 88.0±1.8     | 88.3±1.3            | 88.9±1.1           | 88.9±0.5($\beta$ = 0.1)  |
> | Reddit (ratio = 0.05%)     | 89.6±0.7     | 89.2±0.8            | 89.9±0.9           | 90.1±0.8($\beta$ = 0.45) |
> | Reddit (ratio = 0.50%)     | 90.1±0.5     | 90.1±0.7            | 90.6±0.3           | 91.3±0.7($\beta$ = 0.45) |

---

> ### Author Response · Authors · 2023-11-14
> **Response to Reviewer C2Qb(4/4)**
>
> **Q3: Some of the compared methods, such as GCond, had their code updated recently. Has the performance of these methods been re-evaluated, specifically regarding the results presented in Table 1?**
>
> **A5:**
>
> Thanks for the reviewer’s suggestion.
>
> In brief, we conducted a reproduction using the latest codebase, and the reproduced results **fall within the margin of error** proposed in the original paper.
>
> We replicated the experiments using the latest publicly released code and corresponding commands for GCond and evaluated the final saved synthetic graphs. The results are presented below, and we believe that these results fall within the error range stated in the paper. To prove we do not ignore **the** **use of MSE distance** for gradient matching as mentioned in the code repository. We also conducted experiments under equivalent settings.
>
> Note that all results in the table are averaged over three repetitions for accuracy and consistency.
>
> |                            | GCond(paper) | GCond(reproduction) | GCond(mse) | GCond(norm) |
> | - | - | - | - | - |
> | Cora (ratio = 1.30%)       | 79.8±1.3     | 79.9±1.1            | 78.4±0.8   | 79.2±0.4    |
> | Cora (ratio = 2.60%)       | 80.1±0.6     | 80.0±0.3            | 72.4±0.5   | 80.8±0.4    |
> | Cora (ratio = 5.20%)       | 79.3±0.3     | 79.1±0.2            | 74.2±0.1   | 79.3±0.6    |
> | Citeseer (ratio = 0.90%)   | 70.5±0.5     | 70.7±0.3            | 64.8±0.5   | 71.7±0.3    |
> | Citeseer (ratio = 1.80%)   | 70.6±0.9     | 70.9±1.0            | 62.3±1.0   | 72.5±0.7    |
> | Citeseer (ratio = 3.60%)   | 69.8±1.4     | 70.3±1.1            | 63.9±0.8   | 72.7±0.4    |
> | Flickr (ratio = 0.10%)     | 46.5±0.4     | 46.6±0.3            | 46.3±0.3   | 45.9±0.2    |
> | Flickr (ratio = 0.50%)     | 47.1±0.1     | 47.1±0.3            | 46.7±0.2   | 46.8±0.1    |
> | Flickr (ratio = 1.0%)      | 47.1±0.1     | 47.1±0.1            | 46.1±0.1   | 47.0±0.1    |
> | Ogbn-XRT (ratio = 0.05%)   | -            | 55.3±0.4            | 49.4±0.3   | 53.2±0.3    |
> | Ogbn-XRT (ratio = 0.25%)   | -            | 68.2±0.7            | 60.7±0.7   | 66.5±0.4    |
> | Ogbn-XRT (ratio = 0.50%)   | -            | 69.1±0.6            | 63.3±0.4   | 68.8±0.3    |
> | Ogbn-arxiv (ratio = 0.05%) | 59.2±1.1     | 60.3±0.6            | 55.3±0.4   | 58.7±0.5    |
> | Ogbn-arxiv (ratio = 0.25%) | 63.2±0.3     | 63.2±0.4            | 62.7±0.7   | 64.4±0.8    |
> | Ogbn-arxiv (ratio = 0.50%) | 64.0±1.4     | 64.9±0.3            | 63.4±0.5   | 64.9±0.4    |
> | Reddit (ratio = 0.01%)     | 88.0±1.8     | 88.3±1.3            | 80.9±1.0   | 85.6±0.6    |
> | Reddit (ratio = 0.05%)     | 89.6±0.7     | 89.2±0.8            | 83.9±0.9   | 87.1±0.7    |
> | Reddit (ratio = 0.50%)     | 90.1±0.5     | 90.1±0.7            | 86.6±0.9   | 88.3±0.6    |
>
>
>
> **Q4: Minor Typo**
>
> **A6:**
>
> Thanks for the comment.
>
> We have carefully reviewed the entire manuscript and corrected all typos. Thank you again for your meticulous review and comments!
>
> ---
>
> [A] Zixuan Jiang et al. Delving into effective gradient matching for dataset condensation. IEEE International Conference on Omni-layer Intelligent Systems (COINS).  2023.
>
> [B] Kim et al. Dataset condensation via efficient synthetic-data parameterization. ICML 2022.
>
> [C] Yixin Liu et. al Towards Unsupervised Deep Graph Structure Learning. WWW 2022.

---

> ### Author Response · Authors · 2023-11-17
>
> Thank you for your valuable suggestions, which we have thoroughly considered and implemented. If there are any additional points you would like us to address or if you have further inquiries, we welcome the opportunity to conduct additional experiments. We anticipate your feedback eagerly.

---

> ### Author Response · Authors · 2023-11-21
> **Official Comment by Authors**
>
> We greatly value the constructive suggestions you have offered. If there are any residual uncertainties or if you have specific areas in mind where additional experimentation could be beneficial, we are fully committed to addressing them. Your ongoing involvement is integral to the refinement of our work, and we are eager to engage in further discussions based on your feedback.

---

> > ### Comment · Reviewer_C2Qb · 2023-11-22
> > **Response to author rebuttal**
> >
> > I appreciate the author response and revision, which resolved some of my concerns. However, I still think the overall novelty of the proposed method is limited, so I keep my original score of 5.

---

### Author Response · Authors · 2023-11-19
**General response**

We thank all the reviewers for their suggestions.

To facilitate the review process, we have provided a summary of the response content at the beginning of the main text for the majority of our replies. Detailed discussions and experimental results follow thereafter.

---

### Meta-Review · Area_Chair_XcKY · 2023-12-06

**Metareview:**

The reviewers agree that the paper presents an interesting approach. The main strength of the paper is the extensive set of experiments, which has been enriched during the rebuttal. Yet, there is also a consensus among reviewers that the approach has limited novelty and originality, which hasn't been resolved during the rebuttal: The authors clarified that they came to the idea of euclidian and cosine distances from a fresh perspective, but if the reviewers and I agree that the arguments they provide are more at the intuitive level than a strong theoretical foundation. While the paper is overall borderline, considering the competitiveness of ICLR, this argumentation is not strong enough to pass the acceptance bar the paper.

**Justification For Why Not Higher Score:**

The approach overall looks like incremental compared to what already exists. In order to be accepted, new, stronger theoretical arguments justifying the specific approach of the authors would likely have increased the score.

**Justification For Why Not Lower Score:**

N/A

---

### Decision · Program_Chairs · 2024-01-16

Reject